

# Spatial and temporal variability of mode-1 and mode-2 internal solitary waves from MODIS/TERRA sun glint off the Amazon shelf.

Carina Regina de Macedo[1,2], Ariane Koch-Larrouy[2], José Carlos Bastos da Silva[3,4], Jorge Manuel Magalhães[3,5], Carlos Alessandre Domingos Lentini[6,7,8], Trung Kien Tran[1], Marcelo Caetano Barreto Rosa[7], and Vincent Vantrepotte[1]

[1]Univ. Lille, CNRS, Univ. Littoral Côte d'Opale, UMR 8187 - LOG - Laboratoire d'Océanologie et de Géosciences, F-59000 Lille, France
[2]LEGOS, Université de Toulouse, CNES, CNRS, IRD, UPS, Toulouse, France
[3]Department of Geosciences, Environment and Spatial Planning, Faculdade de Ciências da Universidade do Porto, Rua do Campo Alegre 687, 4169-007, Porto, Portugal
[4]Instituto de Ciências da Terra, Polo Porto, Universidade do Porto, Rua do Campo Alegre 687, 4169-007, Porto, Portugal
[5]CIIMAR, Universidade do Porto, Rua dos Bragas 289, 4050-123, Porto, Portugal
[6]Department of Earth and Environment Physics, Physics Institute, Ondina Campus, Federal University of Bahia—UFBA, Salvador, Bahia, Brazil
[7]Department of Oceanography, Geosciences Institute, Campus Ondina, Federal University of Bahia —UFBA, Salvador, Bahia, Brazil
[8]Interdisciplinary Center for Energy and Environment (CIEnAm), Federal University of Bahia UFBA, Salvador, Bahia, Brazil

**Correspondence:** Carina Regina de Macedo (carina.macedo@fc.up.pt)

**Abstract.** The Amazon shelf is a key region for intense internal tides (ITs) and nonlinear internal solitary waves (ISWs) generation associated with them. The region shows well-marked seasonal variability (boreal Summer/fall ASOND vs spring MAMJJ) of the circulation and stratification, which can both induce changes in the ISWs physical characteristics. The description of the seasonal and neap-spring tidal variability of the ISWs off the Amazon shelf is performed for the first time using a meaningful

data set composed of more than a hundred MODIS/TERRA imagery from 2005 to 2021, where more than 500 ISW signatures were identified in the sun glint region. Previous studies have documented the existence of mode-1 ISW, but the region appears as a newly described hotspot for mode-2 ISWs. ISWs packets separated by typical mode-1 (95 - 170 km; 2.1 - 3.8 m.s$^{-1}$) and mode-2 (46 - 85 km; 1.0 - 1.9 m.s$^{-1}$) ITs wavelengths have been identified and mapped coming from sites A, B, and F. Site A likely shows a higher ISW activity, after two patches of reflection (first - 150 km from IT generation point on the shelf break,

second - 260 km from shelf break), because waves emanating from site D are focusing on the same propagation path (third patch - 380 km from the shelf break). Patches of higher occurrence of ISWs appear separated by typical mode-1 wavelength likely corresponding to the reflection beams at the surface. A fourth patch structured as a tail with finer scales might indicate some region of instability, a transfer to higher modes or dissipation. The range and values of mode-1 and mode-2 propagation velocities/wavelengths do not show significant differences according to areas A and B. Mode-2/mode-1 ratio is larger for site

B likely linked to shallower pycnocline with higher maximum values when compared to area A. The wave activity is higher during spring tides than neap tides (for both A and B sites). During ASOND, mode-1 ISWs from A exhibit higher wave propagation velocities/wavelengths than MAMJJ. In contrast, no seasonal variation of mode-2 propagation velocities/wavelengths





was found. During ASOND in area A, the reinforcement of the North Equatorial Counter Current appears to play a role in deviating the waves towards the northeast, increasing their phase velocities and their eastern traveling direction component

which gives them an extra offshore acceleration. The impact of the circulation on the propagation velocities/wavelength is even more evident for the shorter-scale waves. During ASOND, when the circulation has higher small-scale variability the ISWs propagate in a wider pathway and have a higher diversity of propagation velocities. Calculations of the IT velocities using the Taylor-Goldstein equation supported our results of the presence of mode-2 ISWs associated with mode-2 IT wavelengths in the study area and additionally into the ISW/IT seasonal variability in terms of waves with higher diversity and higher mean

values of wavelength during ASOND.

## 1   Introduction

Nonlinear internal solitary waves (ISWs) are generated in the ocean by various processes, including the interaction of the flow with underwater sills/banks, and the evolution/disintegration of internal tides (ITs) (Jackson et al., 2012; Alford et al., 2015).

Their turbulent mixing and strong horizontal and vertical currents have an impact on oceanic physical and biological processes (e. g., redistribution of heat, and momentum across oceanic basins, and nutrients supply for photosynthesis) (Sandstrom and Elliott, 1984; Huthnance, 1995; Munk and Wunsch, 1998; Muacho et al., 2013) while ISW can also represent a source of hazards for economic activities (e. g., aquaculture and offshore drilling operations) (Osborne et al., 1978; Hyder et al., 2005).

The Amazon shelf has been reported in the literature as being an important hotspot for intense internal tides (IT) and internal

solitary waves (ISWs) generation (Brandt et al., 2002; Magalhães et al., 2016; Lentini et al., 2016; Bai et al., 2021; Tchilibou et al., 2022). Different works have already documented the presence of ISWs in the Amazon Shelf with studies illustrating their propagation both offshore (Brandt et al., 2002; Magalhães et al., 2016) and along the continental shelf (Lentini et al., 2016; Bai et al., 2021). The former are associated with IT hotspots generated over the steep slopes of the Amazon shelf break and disintegrate into short-scale waves several hundred kilometers from the shelf break (Magalhães et al., 2016). The shorter-scale

ISWs are trapped in the IT troughs, both propagating together (Jackson et al., 2012). Magalhães et al. (2016) identified two regions (called A and B) as being the most energetic generation sites (see their Fig. 1). In Tchilibou et al. (2022), more than 6 sites of internal tide generation were identified along with the Amazon shelf break, A and B remaining the strongest.

Intra-seasonal to seasonal variability of the circulation and stratification and neap-spring tidal forcing are linked to changes in the IT and ISW propagation direction, intensity, wavelength, and, consequently, their velocities (Vlasenko et al., 2012;

Magalhães et al., 2016; Liu and D'Sa, 2019; Tchilibou et al., 2022). The Amazon shelf is characterized by two seasons with well-marked differences in water stratification, surface currents, and mesoscale circulation. In boreal spring (from March to July, hereafter MAMJJ), the currents and mesoscale activity are weaker and the pycnocline is shallower, slightly stronger, and horizontally more homogeneous; during the boreal summer/fall (from August to December, hereafter ASOND), the currents





and mesoscale activity are intensified and the pycnocline is deeper, slightly weaker and has a stronger horizontal gradient
along with the North Brazil Current retroflection/North Equatorial Countercurrent (NBCR/NECC) path (Richardson and Walsh,
1986; Richardson et al., 1994; Silva et al., 2005; Aguedjou et al., 2019; Tchilibou et al., 2022). Seasonal variability of the ISWs
in the region was linked to the seasonality of the NECC, which was pointed to as the mechanism responsible for refracting
the waves toward the northeast and enhancing their velocities during the ASOND time period (Magalhães et al., 2016). The
seasonality of the pycnocline depth and strength was linked in the Amazon shelf to changes in the IT baroclinic mode and
wavelength (Barbot et al., 2021; Tchilibou et al., 2022). Finally, the currents may interact with the IT field creating some
refraction, branching, or even dissipation of the tidal baroclinic flux (Dunphy et al., 2017; Tchilibou et al., 2022). During
ASOND, Tchilibou et al. (2022), in a realistic regional modeling configuration showed that the eddy kinetic energy is higher,
and the mesoscale currents create a more energetic noncoherent baroclinic flux. This study further illustrated that the latter flux
has numerous branching and deviations, and the internal tides field seems more diffuse. The impact of the water stratification
and the seasonal variability was also discussed for mode 2 ITs off the Amazon shelf in Barbot et al. (2021); Tchilibou et al.
(2022). A deeper pycnocline seems to shift mode 2 ITs toward intermediate water layers, decreasing the IT elevation amplitude
and increasing its horizontal surface wavelength (with a lower impact on mode-2 than on mode-1) (Barbot et al., 2021).
During MAMJJ, the shallower and slightly strong pycnocline seems to enhance the generation of higher baroclinic modes ITs,
enhancing the local dissipation (Tchilibou et al., 2022).

In the study area, Magalhães et al. (2016) found ISWs with an average inter-packet distance with typical wavelengths of
long (semi-diurnal) ITs of the fundamental mode (i.e., mode-1 ITs). However, the presence of small-scale ISWs with average
inter-packet distance with a typical wavelength of mode-2 ITs was briefly reported in the region by da Silva et al. (2016). The
authors denominated these smaller-scale features as wave tails. Signatures of small-scale ISWs trailing larger ISWs have been
documented in the South China Sea, Mascarene Ridge of the Indian Ocean, and Andaman Sea (Guo et al., 2012; da Silva
et al., 2015; Magalhães and Da Silva, 2018). In the South China Sea, simulations showed two different processes leading to
short internal waves riding on mode-2 ISW and following a strong mode-1 ISW. The first one is related to the disintegration
of a baroclinic bore, which is generated by the interaction between topography and tidal current. The second process calls for
nonlinear interaction between mode-1 and mode-2 ISWs, which takes place when a faster mode-1 wave overtakes a mode-
2 ISW generated one tidal cycle earlier (Guo et al., 2012). In Mascarene Ridge and the Andaman Sea, the impact of the
IW beam with the pycnocline is pointed to as the mechanism responsible for the generation of mode-2 ISWs subsequently
developing shorter-scale waves (wave tails with mode-1 structure) on its background (da Silva et al., 2015; Magalhães and
Da Silva, 2018). Potential mechanisms for the generation of mode-2 waves have been illustrated and they include the instability
of shoaling mode-1 waves (Helfrich and Melville, 1986) and their interaction with localized sills (Lamb and Warn-Varnas,
2015), propagation of mode-1 waves into horizontally varying stratification regime (Liang et al., 2018) and shoaling mode-
2 semi-diurnal internal tide (Liang and Li, 2019). Although the mode-2 waves are not as fast and energetic as the larger
depression/elevation mode-1 waves, they potentially play a significant role in mixing and enhancing the fluxes of nutrients and
heat in the water column because of their location in the middle of the pycnocline (Moum et al., 2008). The mode-2 waves
receive less attention in the literature compared to mode-1 although some works are documenting mode-2 waves propagating



with mode-1 tail in Knight Inlet, British Columbia (Farmer and Smith, 1980) and or illustrating their occurrence following

mode-1 waves in the South China Sea (Yang et al., 2009; Liu et al., 2013).

Remote sensing (RS) is a key observation tool for providing new insights into the ISW generation, propagation, and dissipation mechanisms. Research efforts concerning ISWs are often based on synthetic aperture radar, SAR, and on optical images acquired under sun glint conditions (i. e., areas where the sunlight specular or near-specular reflects directly to the sensor viewing angle) (Jackson and Alpers, 2010; da Silva et al., 2011; Liu et al., 2014; Magalhães et al., 2016). Signatures of oceanic

features on sun glint imagery are produced by variations of short-scale sea surface roughness which cause changes in the image glitter brightness (Jackson and Alpers, 2010; Kudryavtsev et al., 2012). Since ISWs produce leading bands of rough followed by smooth sea surface roughness associated, respectively, with convergent and divergent surface currents, this oceanic feature can be observed in sun glint imagery (Alpers, 1985; Jackson and Alpers, 2010).

Here, we study the ISWs off the Amazon shelf using for the first time a comprehensive data set composed of more than

a hundred images acquired by the Moderate Resolution Imaging Spectroradiometer (MODIS) onboard the TERRA satellite (Jan-2005 to Dec-2021). ISWs with inter-packet distance with typical wavelengths of mode-1 ITs and short-scale ISWs (wave tails with inter-packet distance with typical wavelengths of mode-2 ITs) have been mapped and their propagation velocities and directions are analyzed, considering their seasonal and neap-spring tidal variability. Calculations of the IT phase velocities using the Taylor-Goldstein equation (TGE) supported our results of the presence of shorter-scale ISWs tails separated by mode-

2 IT wavelengths in the study area and additionally into the ISW/IT seasonal variability. For the first time, the Amazon shelf is described as an important hotspot for shorter-scale ISWs coupled with mode-2 ITs.

## 2 Data and Methods

### 2.1 Remote sensing and reanalysis data

The RS data set is composed of 140 images (acquired from 01 January 2005 to 31 December 2021) of Level 1B data

from the MODIS sensor onboard the TERRA satellite. The images were acquired off the Amazon shelf where the presence of ISW signatures was identified in the sun glint region using the Band 6 centered at 1640 nm with a spatial resolution of 500 meters. Level 1B MODIS/TERRA images were collected from NASA's Earth Science Data System, ESDS (https://earthdata.nasa.gov/). The cloud coverage (especially during the months of MAMJJ) and the position of the sun glint area, which changes its position over the year, are limiting factors for our samples. The Global Ocean Ensemble Physics Re-

analysis (EPR) data provides a 3D-gridded description of the global oceanic physical state at 0.25-degree resolution, starting from January 1993 until December 2019. The data is produced by Mercator Ocean International as part of the Copernicus Programme (https://marine.copernicus.eu/), using a multi-numerical ocean model (GLORYS2V4 from Mercator Ocean, France, ORAS5 from ECMWF, GloSea5 from Met Office, United Kingdom, and C-GLORSv7 from CMCC, Italy) ensemble approach and data assimilation of satellite and *in situ* observations. The daily mean average of temperature, salinity, and currents vari-

ables was acquired from 2005 to 2019 for 75 vertical levels.



## 2.2 Remote sensing processing

The ISW signatures were visually identified and manually extracted for each MODIS/TERRA scene of our data set. Signatures of non-linear ISWs can be visualized as leading bands of increased sea surface roughness followed by bands of decreased roughness (Alpers, 1985; Jackson and Alpers, 2010). The leading wave of each ISW packet was mapped for each image of our

data set and the distance between two consecutive leading wave signatures (inter-packet distance, hereafter called wavelength since it is associated with typical IT wavelengths) was calculated considering the vector which connects the middle point of each consecutive ISW signatures, perpendicular to the ISW crests. An image showing a typical view of this study region in which it can be seen that ISW signatures are often found with typical mode-1 IT wavelengths but also those of mode-2 IT (hereafter called mode-1 and mode-2 internal waves) can be seen in Figure 1. The average wave propagation velocity was

calculated considering the period of the semi-diurnal IT of 12.42 hours. The ISW propagation direction, $pd$, was automatically retrieved from the RS data considering the angle between the North and the direction of the vector which connects the middle point of two consecutive packets (in a clockwise direction), i. e., $pd = 0°$ means ISWs propagating from the South to the North and $pd = 90°$ means propagating from West to East.

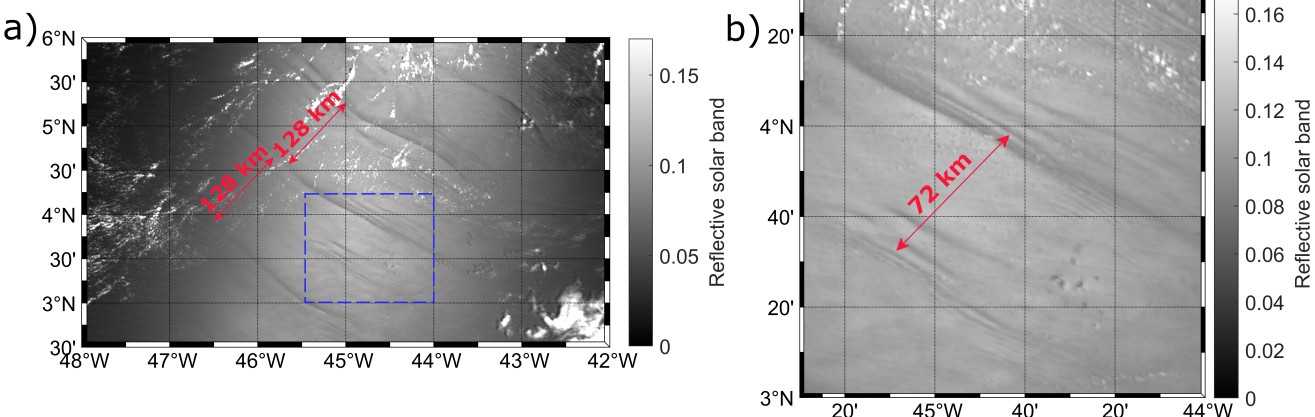

**Figure 1.** Level 1B MODIS/TERRA image, band 6, acquired on 10th October 2014 shows: (a) a typical view of this study region in which it can be seen that ISW signatures are often found with typical mode-1 IT wavelengths. The blue rectangle represents the area where (b) signatures associated with mode-2 ITs are found.

## 2.3 Theoretical calculation of IT velocities

This subsection describes the numerical method proposed by Lian et al. (2020) to solve the viscous TGE for both instabilities and waves in order to calculate the ITs propagation velocities. The equation is used to explain the existence of mode-2 IT waves off the Amazon shelf, including the understanding of the wave's seasonal and near-spring tide variability in terms of shear and water stratification.





Considering a horizontal wave vector $\mathbf{k}$ on a horizontal background flow $\mathbf{U}_h = \{u(z), v(z)\}$, for a small-amplitude and normal mode disturbance, the component of the background flow parallel to the wave vector rules the mode evolution (Lian et al., 2020):

$$U = \frac{\mathbf{k} \cdot \mathbf{U}_h}{k} \tag{1}$$

where $k = \sqrt{k_z^2 + k_m^2}$ is the wave number magnitude, with $k_z$ and $k_m$ being a real wave number with zonal and meridional components. The buoyancy is defined as $b = g(\rho_0 - \rho)/\rho_0$, where g is the gravitational acceleration and $\rho_0$ is the characteristic value of the density $\rho$. The buoyancy perturbations vertical and velocity (i. e., $w'$ and $b'$, respectively) are considered to disturb the parallel shear flow $U(z)$ (Lian et al., 2020).

$$b' = \Re\left(\hat{b}(z) e^{\sigma t + ikx}\right) \tag{2}$$

$$w' = \Re\left(\hat{w}(z) e^{\sigma t + ikx}\right) \tag{3}$$

The terms $\Re$ and $\Im$ refer to the real and imaginary parts, respectively, and $i = \sqrt{-1}$. The complex vertical structure functions of vertical buoyancy and velocity are, respectively, $\hat{b}$ and $\hat{w}$. $\sigma$ is the complex growth rate and $x$ is the horizontal coordinate parallel to the wave vector. The effects of viscosity and diffusivity (respectively, $A_h$, $A_v$ and $K_h$, $K_v$ where the subscripts $h$ and $v$ mean horizontal and vertical components, respectively) are included and their linearized normal-mode equations are (Liu, 2010; Lian et al., 2020):

$$(\sigma + ikU)\nabla^2\hat{w} - ik\frac{d^2 U}{dz^2}\hat{w} = T_w\hat{w} - k^2\hat{b}, \tag{4}$$

$$(\sigma + ikU)\hat{b} + \frac{dB}{dz}\hat{w} = T_b\hat{b} \tag{5}$$

with viscous and diffusive operators

$$T_w = \frac{d^2}{dz^2}\left(A_v\frac{d^2}{dz^2}\right) - k^2\frac{d}{dz}\left[(A_h + A_v)\frac{d}{dz}\right] + k^4 A_h, \tag{6}$$

$$T_b = \frac{d}{dz}\left(K_v\frac{d}{dz}\right) - k^2 K_h \tag{7}$$





Thus, we can deal with the viscous TGE (Equations 4 - 7) as a generalized differential eigenvalue problem:

$$
155 \quad \sigma \begin{pmatrix} \nabla^2 & \mathbf{0} \\ \mathbf{0} & \mathbf{I} \end{pmatrix} \begin{pmatrix} \hat{w} \\ \hat{b} \end{pmatrix} = \begin{pmatrix} -ikU\nabla^2 + ik\frac{d^2U}{dz^2} + T_w & -k^2 \\ -\frac{dB}{dz} & -ikU + T_b \end{pmatrix} \begin{pmatrix} \hat{w} \\ \hat{b} \end{pmatrix} \tag{8}
$$

where $\mathbf{0}$ and $\mathbf{I}$ are the zero and identity matrices, respectively. Considering, $\nabla^2 = d^2/dz^2 - k^2$, the boundary conditions as $\hat{w} = 0$ (impermeable) and $\hat{b} = 0$ (constant buoyancy).

The phase speed of internal gravity waves is described as (Smyth et al., 2011; Lian et al., 2020):

$$
c = \frac{w}{k} \tag{9}
$$

where $w = -\Im(\sigma)$. It is worth noticing that the phase speed is depicted complex number, whose imaginary part is considered zero for stable wave solutions and is the eigenvalue of Equation 8. The underestimation of the ISW phase velocity calculated using the TGE is expected since the equation calculates linear wave phase speed and the nonlinear effects increase the speed of the linear waves (Alford et al., 2010; da Silva et al., 2011). The nonlinear phase speed can be corrected using the Korteweg–De Vries (KdV) equation (Hammack and Segur, 1974; Alford et al., 2010), however, this theory applies only to shallow waters with uniform depths which is not the case in our study area.

The waves' velocities of all modes are calculated using Equations 8 and 9. The local values of stratification and shear were taken from daily and monthly reanalysis data for each location where ISWs were identified considering the entire period of time (from 2005 to 2019). The current velocities were decomposed in the ISW traveling direction. Here, positive velocity means current flowing in the same direction as the ISWs/IT; while a negative one means current flowing in the opposite direction. The separation between the different wave modes is based on the probability distribution of the velocities predicted by the viscous TGE for mode-1 and mode-2, considering the monthly reanalysis data.

## 2.4 Statistical analysis

The normality of the distribution associated with each considered parameter (ISW wavelength/velocity) was evaluated using the Shapiro-Wilk test, SWT. The comparison of the mean of the different groups of data considered here was then performed using a parametric test (Student t-test) when the distribution of the sample was following a normal distribution; while the non-parametric test (Mann-Whitney-Wilcoxon Test, MWWT) was applied when this condition was not valid or for comparison of unbalanced size groups (number of samples in one group more than 3 times the number of samples of the other one). The non-parametric Kruskal-Wallis H test (KWT) was performed to determine if there are statistical differences among more than two independent groups. The non-parametric kernel density (KD) estimation was used for probability density estimation when a parametric distribution could not properly describe our variable.



## 3 Results

All ISW occurrences identified off the Amazon shelf found by direct examination of images are displayed in Figure 2. The number of occurrences of the waves reveals coherent patches (more than 35 occurrences) of the site A organized in a southwest-northeast axis, following a very similar pattern as the one of the M2 internal tides dissipation described by Tchilibou et al.
(2022). This suggests that when reflecting the surface or subsurface the internal tides get unstable and generate strong ISW. Also sites B and F (respectively, eastern, and western paths) show some events. The distance between the generation point A and B (isobath of 200 m) and the first ISW sub-patch of higher occurence is around 150 and 248 km, respectively. In site A, regions of higher occurrence of ISWs are structured into sub-patches separated between each other by mode-1 typical wavelength (see Table 1) and the sub-patch further northeast is structured as a tail with finer scales.

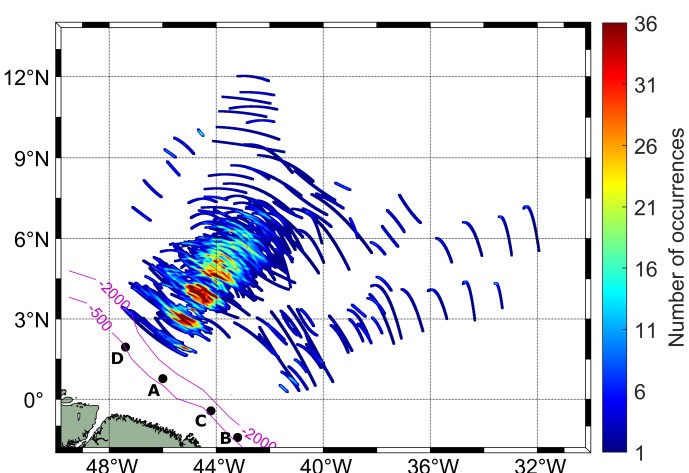

**Figure 2.** Spatial density map of occurrences of ISW signatures visible in MODIS/TERRA images.

First analysis was done considering the ISWs with intra-packet distances associated to mode-1 IT, Figure 3-(a) gives further insight into the horizontal structure of the northeastward-propagating waves, revealing a largely unimodal distribution of crest lengths that are strongly skewed toward the shorter end. These ISW packets are regularly observed to reach crest lengths ranging up to 372 km, although most of the observations are characterized by crest lengths between 70 and 90 km. SAR-derived ISWs are generally not affected by cloud cover on the contrary of sunglint-derived wave identification. This could explain the skewed
distribution here observed toward lower values due to the intense cloud coverage associated with the Inter Tropical Convergence Zone near the Amazon region. The intra-packet distance distribution, which is the distance between waves of the same packet, shows a unimodal distribution shifted to the smallest distances with most of the observations ranging between 7 and 18 km, with a modal value of ∼ 10 km (Figure 3-(b)). The average intra-packet distance for the entire period is 12 km with a standard deviation of 6 km.





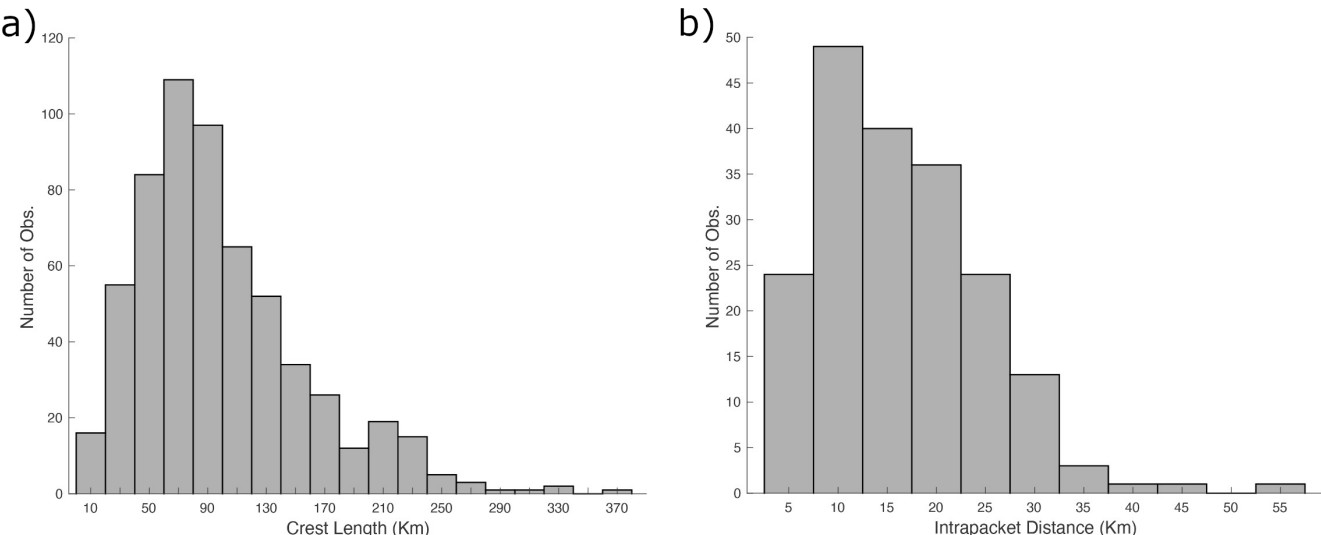

**Figure 3.** (a) ISW crest lengths. The average length is 99 km, with a standard deviation of 58 km. (b) ISW intrapacket distance distribution.

**Table 1.** Distance between the sub-patches of higher occurrence of ISWs in region A. The first sub-patch correspond to the one further Southwest.

| Higher occurrence sub-patches | Distance (km) |
|---|---|
| 1 - 2 | 110 |
| 2 - 3 | 123 |
| 3 - 4 | 115 |

## 3.1 ISWs temporal distribution of first and second baroclinic modes

The monthly and yearly distributions of the number of RS images in which at least one ISW signature was identified (clear image) and the corresponding number of normalized mode-1 and mode-2 wave signatures are presented, respectively, in Figure 4-(a) and (b). Despite the lack of acquisitions for some months, the overall view suggests a seasonal variability in the study area for both mode-1 and mode-2 waves, where the number of wave signatures is more homogeneous during the months of ASOND. The mode-1 waves show a quite homogeneous distribution according to the years; while the number of mode-2 waves has a more evident variation. The number of clear RS images are not equally distributed among the months, most of the images (84%) being identified during the dry season (less cloudy coverage) from August to October. The number of clear images is however more homogeneously distributed from one year to another, despite their significant inter-annual variability. On average, the annual probability of finding a mode-1 signature is more than 4 times the probability of finding a mode-2 one.

Looking now at their horizontal distribution, mode by mode, Figure 5 shows that, ISWs emanate from several IT generation sites mainly from A, B, and F along with the shelf break near the 500 m isobath as previously described by Magalhães et al.



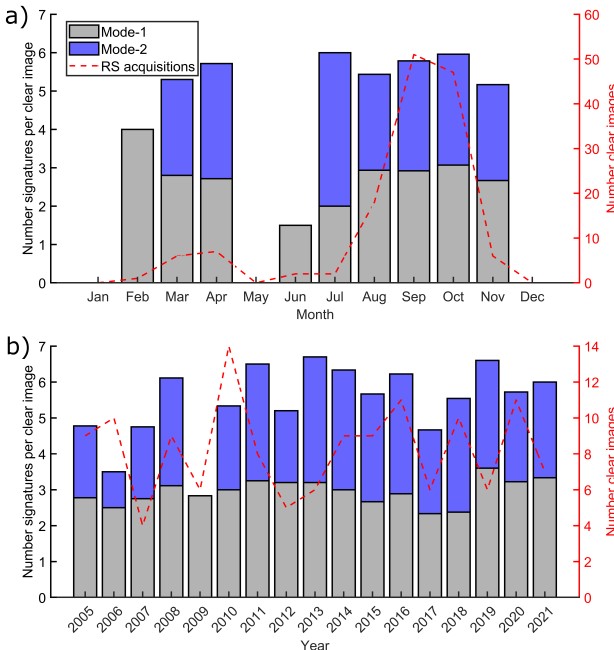

**Figure 4.** The monthly (a) and yearly (b) distributions of the number of RS images in which at least one ISW signature was identified (clear image), dashed red line, and the corresponding number of normalized mode-1 (gray bars) and mode-2 (blue bars) ISW signatures.

(2016); Tchilibou et al. (2022). It is important to point out that in the central path ISW signatures may contain waves from A and D sites. This is more evident considering the mode-2 signatures, where we observe a wave signature emanating from the D site and joining the A path (see a green rectangle in Figure 5-(b)). This example of joining rays of propagation may explain
why the "A" path is the strongest as it focuses rays from D as well. Furthermore, for mode-2 wave signatures in the eastern path, we cannot rule out the presence of signatures from site C. Taking into account the small number of signatures that come from sites F and C, the analysis presented in our paper does not consider those signatures, although a combination of other satellite sensors might help retrieve a stronger signal from these sites (Rosa et al., 2021).

In area A, 353 mode-1 wave signatures and 103 mode-2 ones were found. The wave's propagation velocity/wavelength his-
togram (Figure 6-(a)) follows a bi-modal normal distribution (SWT, $p > 0.05$, see the method for details on the test performed). The groups are associated with waves of different baroclinic modes, and the distribution with a higher mean propagation velocity/wavelength value is likely associated with waves of mode-1 (Table 2). The mean mode-1 and mode-2 wavelength (velocity) deduced from the RS data are, respectively, $131.90 \pm 16.00$ km ($2.94 \pm 0.40$ m.s$^{-1}$) and $70.40 \pm 7.50$ km ($1.57 \pm 0.20$ m.s$^{-1}$). In the area, Magalhães et al. (2016) observed ISWs of fundamental mode propagating with similar mean velocities (i. e.,
$3.1$ m.s$^{-1}$). The KD of the IT propagation velocities calculated using the TGE (Figure 6-(b)) varies according to their mean propagation velocity in a similar pattern which was found in the analysis of the RS scenes, supporting our decision to separate the ISWs according to the different baroclinic modes. The simulated mode-1 mean propagation velocity is underestimated by



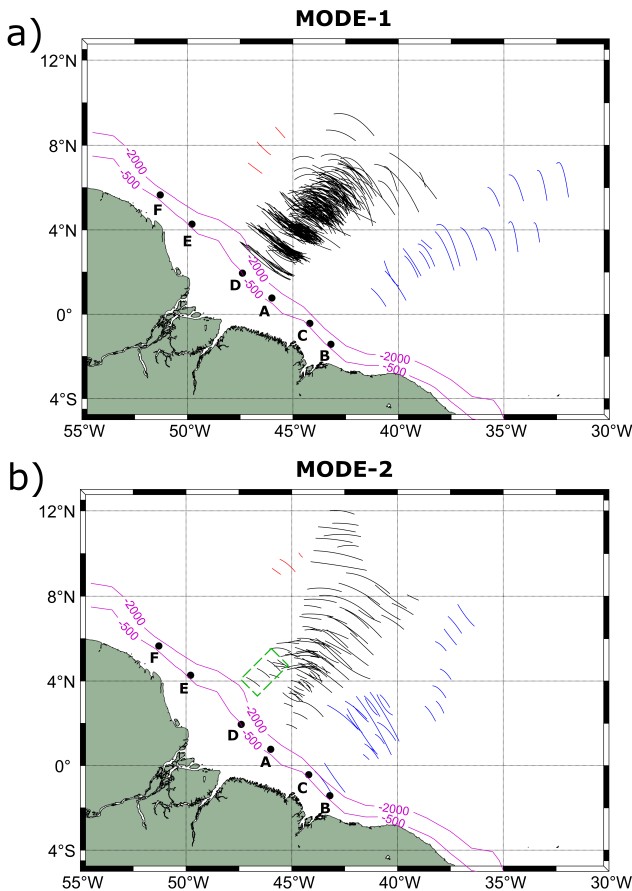

**Figure 5.** (a) mode-1 and (b) mode 2 ISW composite map derived from 140 MODIS/TERRA data acquired under sun glint conditions from 2005 to 2021. Black, blue, and red solid lines correspond to ISW signatures which emanate likely from IT generation points A, B, and F, respectively. All identified signatures were considered and depicted on the map, with a total of 507 signatures among which 377 are associated with mode-1 and 130 correspond to mode-2 internal waves.

∼ 20%, see Table 2. da Silva et al. (2011) found an underestimation of the phase speed calculated by the TGE of ∼ 12% in the Mascarene Plateau considering the ocean depth between 3 - 3.8 km. According to Alford et al. (2010), in the South China Sea
nonlinear waves of M2 frequency travel at a phase speed 1.5 times the linear wave phase speed. The nonlinear phase speed is positively proportional to the surface wave elevation, which explains the higher underestimation of the mode-1 waves (Jeans, 1995; Barbot et al., 2021; Tchilibou et al., 2022).

In area B, 19 mode-1 wave signatures and 26 mode-2 ones have been identified. In contrast with area A, here the amount of mode-2 signatures is, therefore, higher than the amount of mode-1 by 1.3 times. The depth of the pycnocline in the study area
was defined as the depth corresponding to the maximal value of the Brunt–Väisälä frequency. A slightly shallower pycnocline with higher maximum values of the Brunt–Väisälä frequency is found in area B when compared to area A (see Figure 7-(a)),



suggesting that stronger higher mode internal tide generation is expected in area B (Barbot et al., 2021; Tchilibou et al., 2022) in good agreement with our findings.

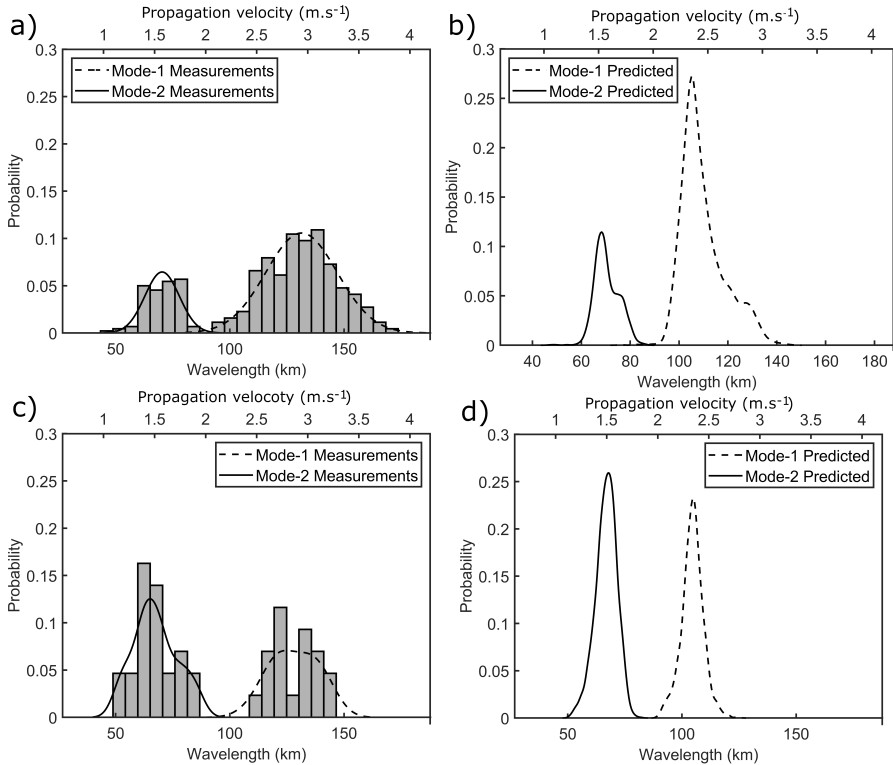

**Figure 6.** Histogram of mode-1 and mode-2 internal waves for (a) area A and (c) area B in gray color. The fitted normal distribution of the ISW propagation velocity/wavelength calculated from the RS data is shown in black. KD of the predicted velocities by solving the viscous Taylor-Goldstein equation using monthly reanalysis data for (b) area A and (d) area B. Mode-1 and mode-2 waves are shown as dotted and continuous lines, respectively.

The histogram of the wave's propagation velocity/wavelength in area B is shown in Figure 6-(c), similarly to area A a bi-
modal distribution is found where two groups vary according to their mean propagation velocity/wavelength, see Table 2. In this case, because of the small number of samples, a KD is estimated. The mean mode-1 and mode-2 wavelength (velocity) deduced from the RS data are, respectively, $128.20 \pm 9.70$ km ($2.87 \pm 0.20$ m.s$^{-1}$) and $69.40 \pm 11.60$ km ($1.55 \pm 0.30$ m.s$^{-1}$). For ISW of the fundamental mode, Magalhães et al. (2016) found waves with similar mean propagation velocity in area B (i. e., 2.7 m.s$^{-1}$). The TGE allows a relevant prediction of the propagation velocity/wavelength distribution of mode-1 and
mode-2 waves, with mode-1 velocities being underestimated by 22% (Figure 6-(d)). Although the mode-1 mean propagation velocity/wavelength is slightly lower in area B compared to area A (about 2.8% lower from the RS analysis and 4.5% from the TGE), the mode-1 and mode-2 mean propagation velocity/wavelength values do not vary significantly according to the different study areas ($p > 0.05$, MWWT).





**Table 2.** Values of IT wavelength and average propagation velocity calculated from RS data and predicted by solving the viscous TGE using monthly reanalysis data in areas A and B according to the different baroclinic modes of the waves.

| Area | Order of baroclinic mode | Data Source | Wavelength (Km) | | Propagation velocity ($m.s^{-1}$) | |
|---|---|---|---|---|---|---|
| | | | Mean ($\pm$ std) | Minimum - maximum | Mean ($\pm$ std) | Minimum - maximum |
| A | 1 | RS | 131.90 ($\pm$ 16.00) | 96.30 - 170.00 | 2.94 ($\pm$ 0.40) | 2.20 - 4.00 |
| | | TGE | 109.69 ($\pm$ 8.98) | 76.36 - 146.18 | 2.45 ($\pm$ 0.20) | 1.71 - 3.27 |
| | 2 | RS | 70.40 ($\pm$ 7.50) | 46.40 - 84.20 | 1.57 ($\pm$ 0.20) | 1.00 - 1.90 |
| | | TGE | 70.41 ($\pm$ 4.98) | 46.44 - 88.80 | 1.57 ($\pm$ 0.11) | 1.04 - 1.99 |
| B | 1 | RS | 1.28.20 ($\pm$ 9.70) | 109.80 - 141.95 | 2.87 ($\pm$ 0.20) | 2.46 - 3.17 |
| | | TGE | 104.61 ($\pm$ 4.82) | 89.13 - 125.54 | 2.34 ($\pm$ 0.11) | 1.99 - 2.81 |
| | 2 | RS | 69.40 ($\pm$ 11.60) | 52.13 - 93.71 | 1.55 ($\pm$ 0.30) | 1.17 - 2.10 |
| | | TGE | 66.85 ($\pm$ 4.71) | 50.54 - 82.32 | 1.50 ($\pm$ 0.11) | 1.13 - 1.84 |

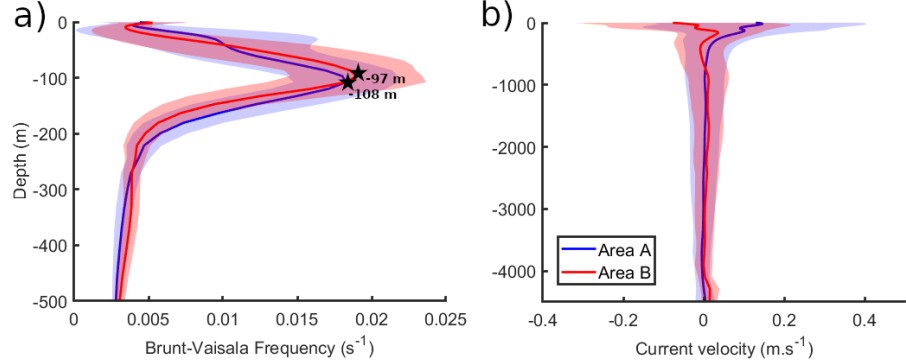

**Figure 7.** (a) Brunt-Väisäla frequency and (b) mean current velocity decomposed along the ISW traveling direction for areas A and B derived from ensemble physics reanalysis data. The bands represent the standard deviation over the period from 2005 to 2019.

The spatial distribution of the mode-1 and mode-2 waves according to their propagation velocities allows the discrimination of two main branches of waves propagating in different directions in area A, where the most eastern branch is associated with higher propagation speed (Figure 8-(a) and (c)). Globally a higher contrast is found for the branches of the mode-2 waves in terms of both velocities and spatial location when compared to mode-1 waves. In area A, an offshore acceleration is also further observed for the most eastern branch of both mode-1 and mode-2 waves (see Figure 8-(b) and (d), where a cross-shore profile was done and the corresponding values of internal wave wavelengths along the profile were retrieved). The acceleration is slightly more pronounced for mode-2 waves with an offshore increment in the propagation velocities of $\sim 18\%$; while the increment for the mode-1 waves is $\sim 15\%$.





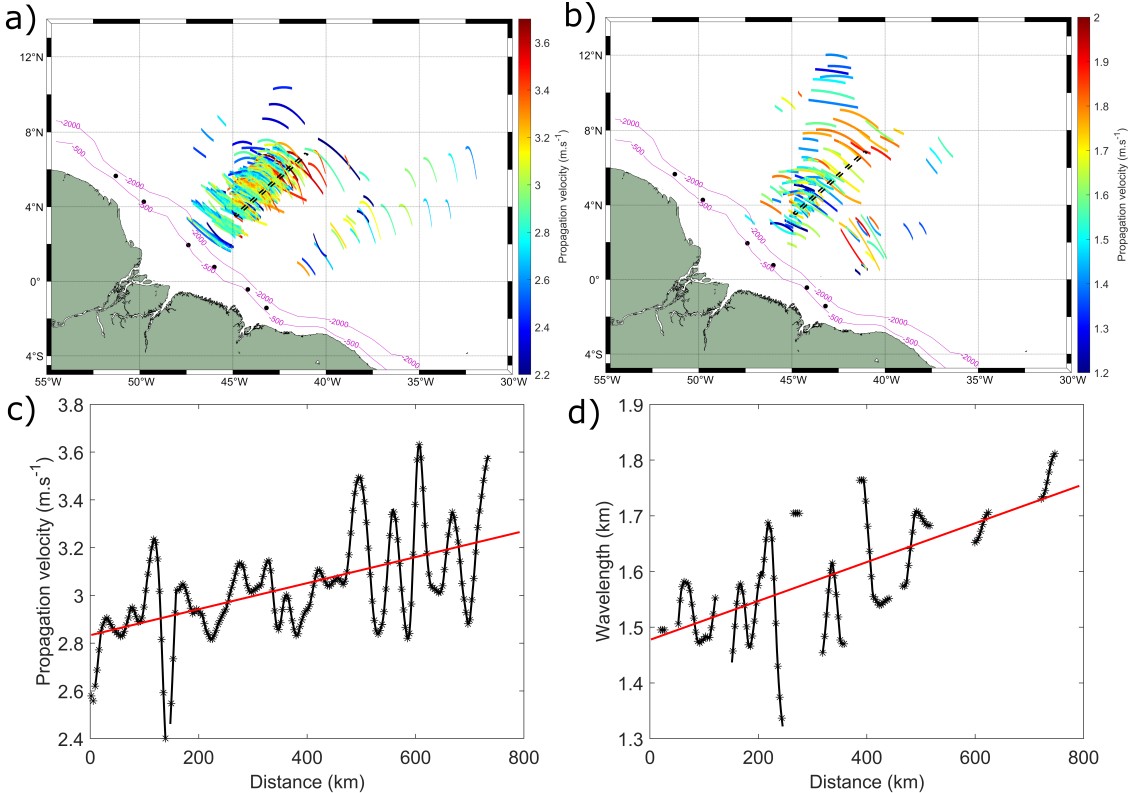

**Figure 8.** ISW propagation directions for (a) area A and (b) area B. ISW propagation angles are clockwise from the North. A $pd = 0°$ denotes ISWs propagating from the South to the North and $pd = 90°$ denotes ISWs from the West to the East.

Besides the analysis of the wavelength and propagation speed, our data set was also used for characterizing the direction of propagation of the ISWs (Figure 9). More than 50% of the mode-1 waves in area A propagate following the path centered at 36°; while mode-2 waves mostly propagate in the paths centered at 36° and 12°. The mode-2 waves propagate in a wider range

of directions when compared to the mode-1 ones. For both mode-1 and mode-2, there are significant differences between the ISW propagation direction paths ($p < 0.01$, KWT, Table 3). The mean velocities/wavelengths tend to increase with the increase of the eastern traveling direction component. This increasing pattern is even more pronounced for the mode-2 waves than for mode-1 (respectively, an increase of 20% and 15%). This suggests that when deviated to the east the waves are accelerated by regional eastward currents.

In area B, mode-1 waves mostly propagate into two different paths centered at 60° and 84° (Figure 9-(b)). The path centered at 36° is not considered in the analysis because of the few samples available. Although we have an increase of the velocity/wavelength with the increase of the northern traveling direction component of 7%, the groups of waves propagating in the two different paths (paths centered at 60° and 84°) do not statistically differ in terms of their velocity/wavelength ($p > 0.05$, MWWT). However, it is important to point out that the lack of samples can impair our analysis. For mode-2, more than 80%




of the waves propagate in the paths centered at $60°$ and $36°$, see Figure 9-(b). The statistical analysis is done considering only the paths centered at $36°$ and $60°$ since few samples are associated with the other paths. In this case, waves for the different propagation paths do not differ statistically in terms of velocity/wavelength ($p > 0.05$, t-test). Area B is less influenced by the NECC than area A. As a matter of fact, current velocities decomposed on the ISW traveling direction are less than half of the respective values found in area A with a higher negative component (i. e., the current flowing in the opposite direction to the waves traveling direction, Figure 7-(b). Compared with the mode-1 waves, the mode-2 ones have a stronger northern component and propagate in a wider pathway in both areas. In area B, mode-1 and mode-2 waves have a more eastern traveling direction component when compared with area A. The stronger dynamic circulation during the boreal spring in area A (Richardson and Walsh, 1986; Richardson et al., 1994; Silva et al., 2005; Aguedjou et al., 2019) appears to significantly act on the wave velocities according to the different paths.

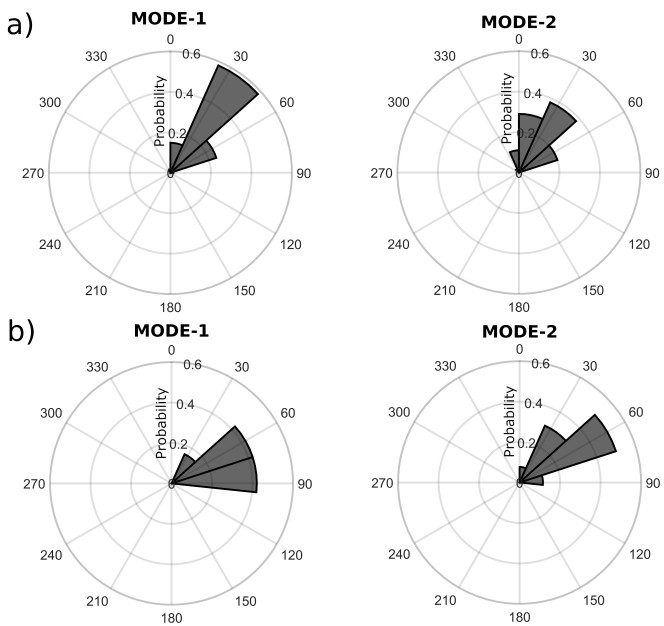

**Figure 9.** Propagation velocities of the (a) mode-1 (b) and mode-2 internal waves off the Amazon shelf. The black dashed rectangle represents the selected cross-shore profile. Cross-shore profile of the ISW propagation velocities for (c) mode-1 and (d) mode-2 waves derived from the RS data. The red line is the fitted linear regression model for the measurements.

### 3.1.1 Spring-neap tidal variability

The occurrence of ISWs according to the tidal conditions (i. e., neap and spring tides) have been investigated for both areas A and B. A more detailed description of wave propagation velocities and directions variability according to the spring-neap cycle was performed on area A. Because of the lack of measurements in area B, the description was not performed in that area.




**Table 3.** Internal tide wavelength and average propagation velocity calculated from RS data in areas A and B according to the different baroclinic mode of the waves and their propagation directions.

| Area | Order of baroclinic mode | Pd path (°) | Mean wavelength (± std) (km) | Mean propagation velocity (± std) ($m.s^{-1}$) |
|---|---|---|---|---|
| A | 1 | 12 | 120.6 (± 14) | 2.7 (± 0.4) |
| | | 36 | 132.4 (± 15.2) | 3.0 (±0.3) |
| | | 60 | 137.8 (± 15.5) | 3.1 (± 0.3) |
| | 2 | 12 | 68.6 (± 9.0) | 1.5 (± 0.2) |
| | | 36 | 71.6 (± 6.6) | 1.6 (± 0.1) |
| | | 60 | 73.9 (± 6.7) | 1.7 (± 0.2) |
| B | 1 | 348 | 64.7 (± 2.8) | 1.4 (± 0.1) |
| | | 36 | 130.1 (± 8.4) | 2.9 (± 0.2) |
| | | 60 | 132.2 (± 11.5) | 3.0 (± 0.3) |
| | | 84 | 123.6 (± 6.7) | 2.8 (± 0.1) |
| | 2 | 12 | 93.7 | 2.1 |
| | | 36 | 67.3 (± 4.8) | 1.51 (± 0.1) |
| | | 60 | 65.1 (± 10.5) | 1.46 (± 0.2) |
| | | 84 | 77.3 (± 10.5) | 1.7 (± 0.2) |

Analysis of the MODIS/TERRA data revealed that the ISW activity is more pronounced near spring tide conditions for both areas (71% of the ISWs signatures for area A and 61% for area B). This result is in line with former studies where higher wave activity near spring tide conditions has been also pointed out (New and Da Silva, 2002; da Silva et al., 2011; Liu and D'Sa, 2019). In both areas, the proportion between mode-2/mode-1 signatures seems to increase from spring to neap tides, in other words, there is more mode 2 waves during neap tides than spring tides. Indeed, in area A, near neap tide, around 28% of the signatures were related to mode-2 waves whereas near spring tide, mode-2 signatures are smaller at 20%. Similarly, in area B, near neap tide, we identified about 61% of the ISW signatures as being related to mode-2 type; while, near spring tide, the mode-2 signatures are smaller with only 48%.

Indeed, for area A, the mode-1 propagation velocity/wavelength values follow a normal distribution for near spring and neap tides (Figure 10-(a)), whose mean values vary according to the different tidal conditions ($p < 0.01$, t-test). The distribution with a higher mean propagation velocity/wavelength is associated with the wave signatures found near neap tide ($137.6 \pm 15.2$ km and $3.1 \pm 0.3$ m.s$^{-1}$); in contrast, near spring tide, the mean propagation velocity/wavelength of the waves decreases about 6% ($129.8 \pm 16.1$ km and $2.9 \pm 0.4$ m.s$^{-1}$). The IT wave propagation velocities calculated using the TGE do not reproduce the





differences between spring and neap tides (the mean predicted value for both spring and neap tides is $109.74 \pm 9.91$ km and $2.46 \pm 0.22$ m.s$^{-1}$; not shown in Figure 10). For the mode-2 waves, no significant differences are found according to the tide conditions ($p > 0.05$, t-test), see Figure 10-(b). The mean wavelength (propagation velocity) for near spring and neap tides are,

respectively, $70.16 \pm 7.59$ km ($1.57 \pm 0.17$ m.s$^{-1}$) and $70.67 \pm 7.55$ km ($1.58 \pm 0.17$ m.s$^{-1}$). No differences are also found according to the tide condition for the velocities predicted by the TGE ($69.00 \pm 5.60$ km and $1.55 \pm 0.13$ m.s$^{-1}$). Further, no significant differences are found in the propagation direction according to the spring-neap tides, for both mode-1 and mode-2 waves (figure not shown).

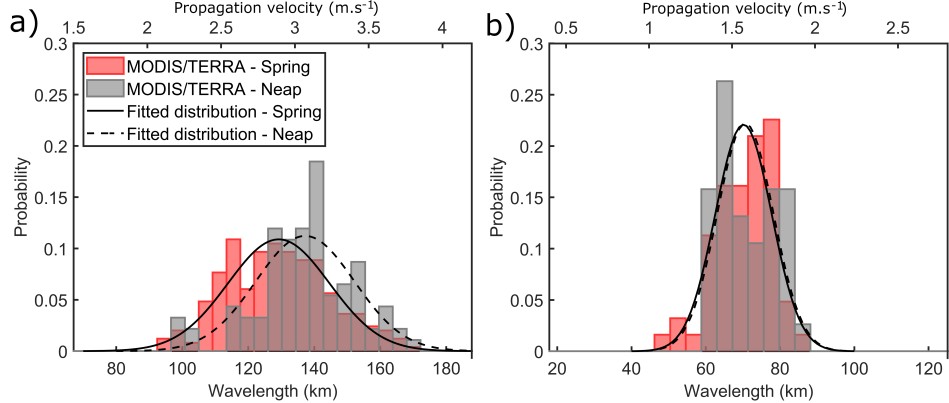

**Figure 10.** Fitted Gaussian distribution of the (a) mode-1 and (b) mode-2 internal wave propagation velocity/wavelength according to the neap (dotted line) and spring (continuous line) tides.

### 3.1.2   Seasonal variability

The seasonal variability of the ISW characteristics in terms of velocity, wavelength, and propagation direction has been further characterized considering the two well-marked seasons on the Amazon shelf, i. e. the boreal spring (from March to July, MAMJJ) and the boreal summer/fall (from August to December, ASOND) following Tchilibou et al. (2022). We only apply this analysis to the A waves, because other sites do not have enough data in both seasons. The mode-1 propagation velocity/wavelength values follow a normal distribution for both seasons (Figure 11-(a)) and there is a statistically significant difference

between the boreal spring and boreal summer/fall ($p < 0.01$, MWWT) with a significantly higher propagation velocity/wavelength during ASOND (Table 4). Note that our samples are unbalanced according to the season, i. e., during ASOND we have about 8 times more samples than for MAMJJ. For mode 1, in ASOND, an increase in the waves' wavelength/velocities is observed compared to MAMJJ (14.3% in the wavelength). The pycnocline during ASOND is slightly deeper when compared to MAMJJ (by 11 m, see Figure 12-(a)). Larger wavelengths (higher velocities) of mode-1 internal tides are expected during

ASOND following previous studies (Liu and D'Sa, 2019; Barbot et al., 2021; Tchilibou et al., 2022). Furthermore, the mean current velocity decomposed on the ISW traveling direction has a stronger positive component during ASOND when compared to MAMJJ (Figure 12-(b)). Hence circulation and stratification probably act constructively to increase the wavelength



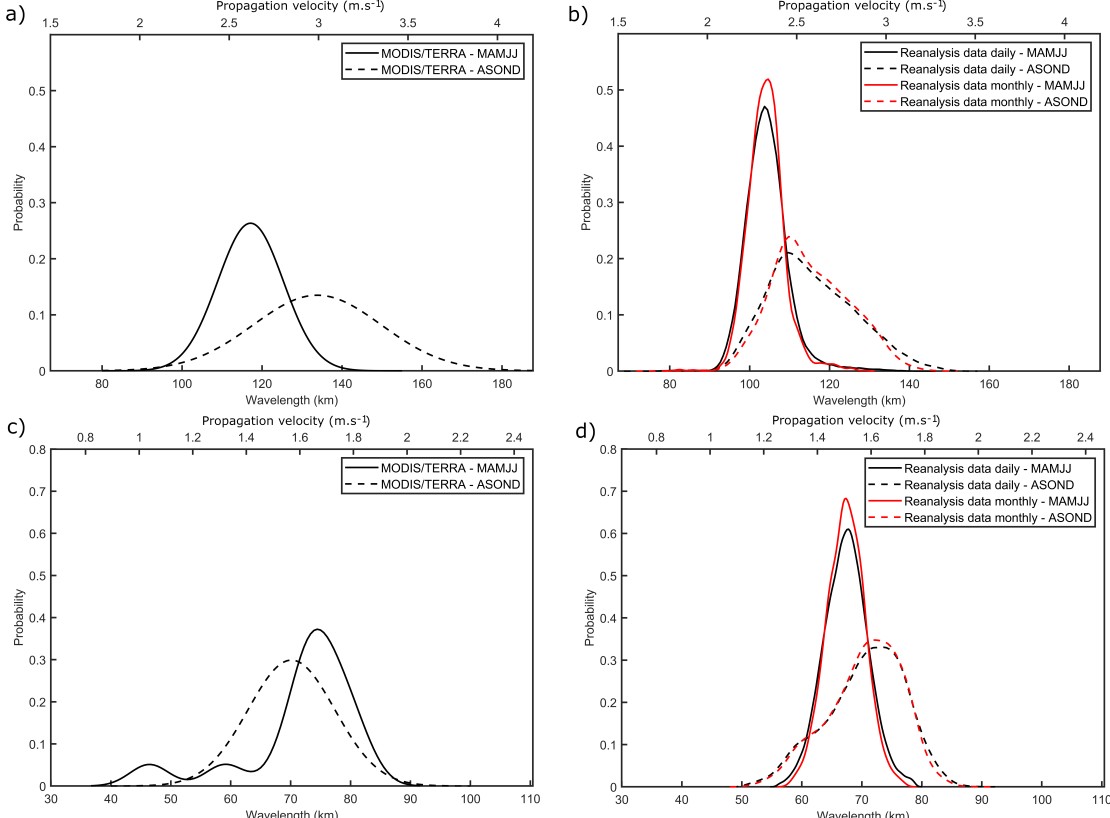

**Figure 11.** Fitted distribution of the ISW propagation velocity/wavelength calculated from the RS data for (a) mode-1 and (c) mode-2 waves for area A. Fitted distribution of the ISW propagation velocity/wavelength predicted by solving the viscous Taylor-Goldstein equation using daily and monthly reanalysis data, respectively, black and red lines for (b) mode-1 and (d) mode-2 waves. MAMJJ and ASOND are shown, respectively, as continuous and dashed lines.

during ASOND, in contrast to MAMJJ. Furthermore, during ASOND, the waves are characterized by a higher diversity in terms of their wavelengths (higher standard deviation) when compared to MAMJJ, suggesting a higher variability of the local

stratification and current shear patterns in the study area during the boreal summer/fall.

The TGE is able to predict the differences in the mode-1 IT velocities/wavelengths between the two seasons ($p < 0.01$, MWWT) with a distribution pattern similar to the one estimated using the RS data (see Figure 12-(b)), i. e., during ASOND the mean waves' velocities/wavelengths and their standard deviation increase when compared to MAMJJ. However, the differences between the two seasons are less evident considering the values predicted by the TGE (mean wavelength value increases by

9.5% during ASOND when compared to MAMJJ). Compared to the RS data, the waves' wavelengths are underestimated by 11% and 14%, respectively, for MAMJJ and ASOND (Table 4). The mean value of velocity/wavelength predicted using daily and monthly reanalysis data are very similar for both seasons. Differences in their standard deviation (standard deviation



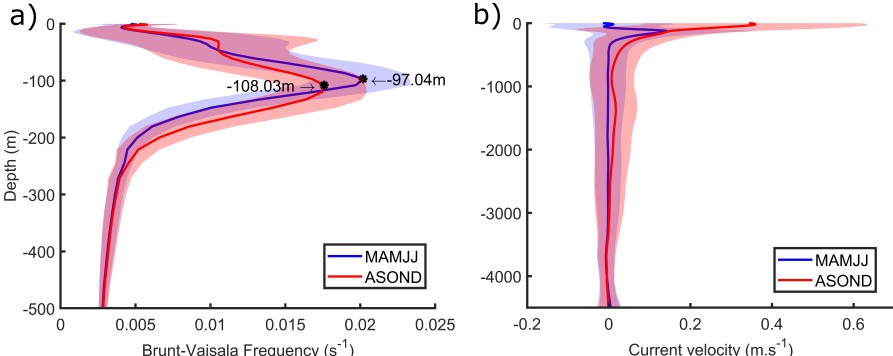

**Figure 12.** Fitted distribution of the ISW propagation velocity/wavelength calculated from the RS data for (a) mode-1 and (c) mode-2 waves for area A. Fitted distribution of the ISW propagation velocity/wavelength predicted by solving the viscous Taylor-Goldstein equation using daily and monthly reanalysis data, respectively, black and red lines for (b) mode-1 and (d) mode-2 waves. MAMJJ and ASOND are shown, respectively, as continuous and dashed lines.

**Table 4.** Values of IT wavelength and average velocity calculated from RS data and predicted by solving the viscous TGE using daily and monthly reanalysis data in areas A and B according to the seasons and the different baroclinic modes of the waves.

| Season | Order of baroclinic mode | Data source | Wavelength (km) | | Propagation velocity ($m.s^{-1}$) | |
|---|---|---|---|---|---|---|
| | | | Mean ($\pm$ std) | Minimum - maximum | Mean ($\pm$ std) | Minimum - Maximum |
| MAMJJ | 1 | RS | 117.1 ($\pm$ 8.2) | 100.16 - 137.84 | 2.62 ($\pm$ 0.18) | 2.24 - 3.08 |
| | | TGE - daily | 104.35 ($\pm$ 5.52) | 74.88 - 145.15 | 2.34 ($\pm$ 0.12) | 1.68 - 3.25 |
| | | TGE - monthly | 104.18 ($\pm$ 4.71) | 80.96 - 128.84 | 2.33 ($\pm$ 0.11) | 1.81 - 2.87 |
| | 2 | RS | 71.83 ($\pm$ 9.47) | 46.43 - 81.89 | 1.61 ($\pm$ 0.21) | 1.03 - 1.83 |
| | | TGE - daily | 67.41 ($\pm$ 3.85) | 45.72 - 88.43 | 1.51 ($\pm$ 0.09) | 1.02 - 1.98 |
| | | TGE - monthly | 67.37 ($\pm$ 3.30) | 56.51 - 84.83 | 1.51 ($\pm$ 0.07) | 1.27 - 1.90 |
| ASOND | 1 | RS | 133.8 ($\pm$ 16.1) | 96.32 - 178.99 | 2.99 ($\pm$ 0.36) | 2.15 - 4 |
| | | TGE - daily | 115.62 ($\pm$ 10.95) | 72.00 - 153.57 | 2.59 ($\pm$ 0.25) | 1.61 - 3.44 |
| | | TGE - monthly | 115.36 ($\pm$ 9.64) | 77.67 - 145.95 | 2.58 ($\pm$ 0.22) | 1.74 - 3.27 |
| | 2 | RS | 70.13 ($\pm$ 7.25) | 53.09 - 84.24 | 1.57 ($\pm$ 0.16) | 1.19 - 1.88 |
| | | TGE - daily | 70.52 ($\pm$ 6.72) | 36.31 - 90.24 | 1.58 ($\pm$ 0.15) | 0.81 - 2.02 |
| | | TGE - monthly | 70.48 ($\pm$ 6.24) | 46.10 - 87.73 | 1.58 ($\pm$ 0.14) | 1.03 - 1.97 |

slightly higher for daily reanalysis data) indicate that using monthly data probably tends to smooth the variability related to stratification and circulation.





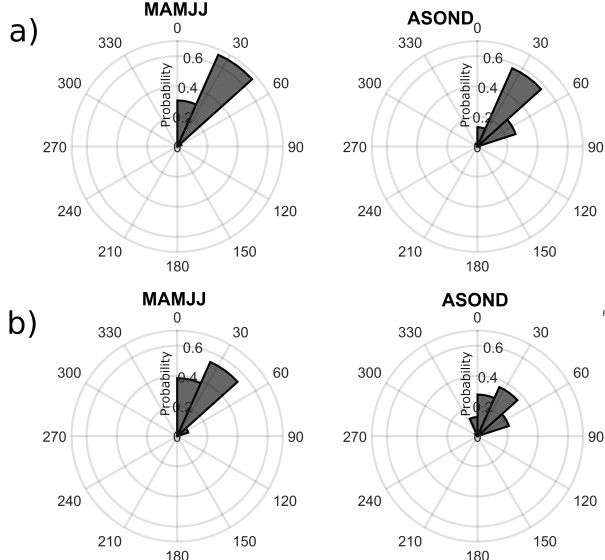

**Figure 13.** ISW propagation directions for (a) mode-1 and (b) mode-2 waves according to the season. ISW propagation angles are clockwise from the North. A pd = 0° denotes ISWs propagating from the South to the North and pd = 90° denotes ISWs from the West to the East.

The mode-2 propagation velocity/wavelength values are non-normally distributed for MAMJJ and normally distributed for ASOND. No variation according to the different seasons is found in contrast to the mode 1 waves ($p > 0.05$, MWWT), see Figure 11-(c) and Table 4. The wave propagation velocities/wavelengths calculated using the TGE vary according to the seasons ($p < 0.01$, MWWT), see Figure 11-(d). During ASOND, the predicted mean velocity increases by 4.6% compared to MAMJJ, and the distribution of the predicted velocities fits quite well with the RS data. The TGE seems to underestimate the

wave propagation velocities by 6.5% during MAMJJ. It is important to point out that the mode-2 signatures identified from the RS observation are not well balanced between the seasons and the period of MAMJJ count with only 13 samples probably impairing our analysis.

During ASOND, the mode-1 and mode-2 waves propagate in a wider direction pathway (Figure 13), and an increase in the wave propagation velocities with increasing eastern traveling direction is found (no differences in the velocities of the waves

are found according to the different pathways during MAMJJ, $p > 0.05$, Mann-Whitney U nonparametric test). The stronger relative differences in the velocities according to the different propagation pathways during ASOND are found for the mode-2 waves, which seem to be more sensitive to changes in the circulation patterns in the study area. In the months of ASOND, the circulation is characterized by the eastward reinforcement of the NECC, which likely plays a role in refracting the waves to the northeast (as pointed as well by Magalhães et al. (2016)) and in increasing their propagation velocities eastward. Furthermore,

as a consequence of the more dynamic mesoscale circulation associated with ASOND, the ISWs seem to spread over the study area during this season (Figure 14), extending the wave penetration further north principally at the end of the summer and early



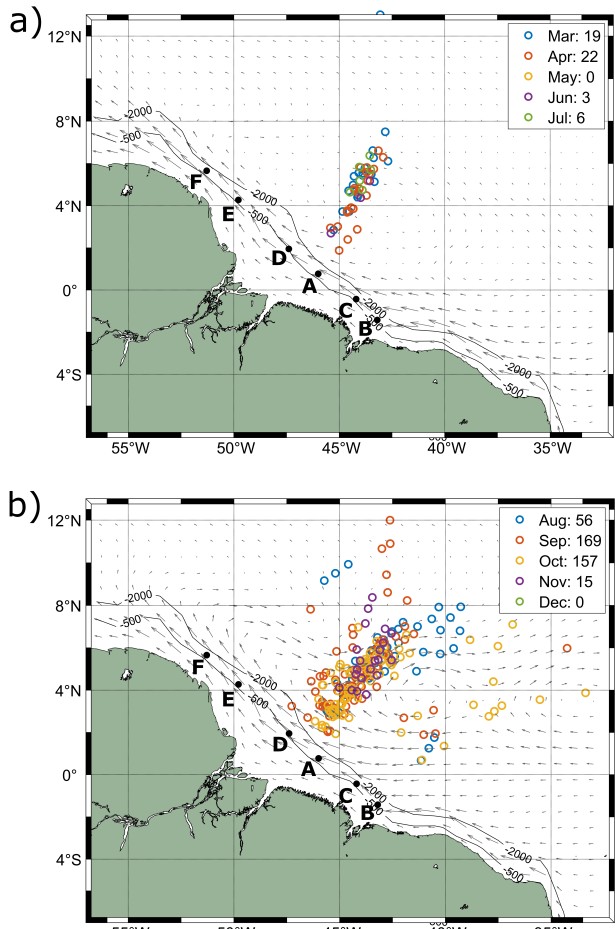

**Figure 14.** Location of the ISW signatures associated with mode-1 and mode-2 internal tides for (a) MAMJJ and (b) ASOND. The colors represent the months when the signatures were identified. The points represent the middle point of each wave signature and the arrows depict the mean surface current speed and direction for each season derived from the EPR data. The number of ISW signatures occurring per month can be found in the legend.

fall (August-October) when maximum values of eddy vorticity are found (Aguedjou et al., 2019). This behavior contrasts with the more straightforward path and lower penetration further north associated with the waves in the months of MAMJJ. Most of the ISW signatures in area A above latitude of $8°$ correspond to mode-2 waves from early September 2014 and 2018. However, 350  it is important to point out that this can be a result of sampling restrictions due to the combination of higher cloudy coverage and the location of the sun glint area during the months of MAMJJ.



## 4 Summary and discussion

This study focuses on the Amazon ISWs occurrence and parameters, such as position, velocity, the direction of propagation, wavelength, and variability at seasonal and spring-neap tidal cycles. The analysis is based on a data set composed of more than 100 MODIS/TERRA images, where more than 500 ISW signatures were identified in the sun glint area.

The central path in Figure 2 has been pointed out in our analysis as the one with stronger ISW activity containing more than 450 signatures. This result agrees with Tchilibou et al. (2022). 2022 who found site A and B with quite similar M2 baroclinic flux horizontal divergence but site A being more efficient in converting the flux into ITs. Another important point is that the central path likely focuses rays emanating from both sites A and D (which is more evident considering the mode-2 wave signatures, see a green rectangle in Figure 5-(b)), which corresponds to the third patch of occurrence from the shelf break, with higher occurrences. In this path, the distance between the generation point A (isobath of 200 m) and the first patch of high occurrence is around 150 km. Regions of higher occurrence of ISWs are structured into sub-patches separated between each other by mode-1 typical wavelength (see Table 1) which might correspond to the reflection beams at the surface. The patch further northeast is structured as a tail with finer scales, being more noisy as the wave get unstable to higher modes. The waves propagate about 350 km without dissipation and then suffer changes in their wavelength which could indicate some instability, a transfer to higher modes, or dissipation. Our finds are suitable with the results presented in Tchilibou et al. (2022).

Previous studies have documented the existence of mode-1 ISW (Magalhães et al., 2016), but in fact, the region appears as a newly described hotspot for mode-2 ISWs. The ISW velocity/wavelength deduced from the RS data showed a bi-modal distribution with two well-separated peaks for both areas A and B, allowing us to separate the ISW associated with two IT baroclinic modes: wavelength (velocity) ranging from 95 - 170 km (2.1 - 3.8 $\text{m.s}^{-1}$) for mode-1 and 46 - 85 km (1.0 - 1.9 $\text{m.s}^{-1}$) for mode-2. Barbot et al. (2021) found using results from ocean modeling horizontal surface wavelengths varying from 110 - 120 km and 70 - 80 km for mode-1 ITs in the study area which is in fair agreement with our results considering that nonlinear effects associated with the ISWs increase the phase speed of the waves and the intraseak variability of current and stratification that explain our more dispersive results compared to the model. Zhang and Li (2022) found a bi-modal distribution of the ISW phase speed in the region as well, calculating using a data-driven machine learning model ISW with a wide range of velocities from values lower than $1\ \text{m.s}^{-1}$ up to $4\ \text{m.s}^{-1}$. Although the underestimation of the mode-1 wavelengths/velocities by the TGE in the study area are of the order of 20-22%, the simulated wavelengths/velocities showed two well-separated distributions for mode-1 and mode-2 waves with a similar pattern to the one deduced from the RS data supporting our decision to split the waves according to their different baroclinic modes.

The range and values of mode-1 and mode-2 propagation velocities/wavelengths do not show significant differences according to areas A and B.

However, area B seems to have a higher proportion of mode-2/mode-1 waves (i. e., stronger higher mode internal tide generation) likely linked to its shallower pycnocline with higher maximum values when compared to area A (see Figure 7). However, we can not rule out the fact that ISWs emanating from site C may be counted as coming from from site B influencing these results since according to Tchilibou et al. (2022) site C is the one most favorable to local dissipation (higher





modes generation connected to a higher probability of instability, thus higher local dissipation). In both areas, neap-spring tidal variability is found, i. e., the wave activity is higher during near spring tides, which is coherent with the larger tidal currents during spring tides. This result is in line with former studies where higher wave activity near spring tide conditions has been also pointed out (da Silva et al., 2011; Liu and D'Sa, 2019). In addition, the proportion of mode-2/mode-1 waves increases

from spring to neap tide conditions.

Seasonal variability of the mode-1 waves was found in area A, where a higher diversity (higher standard deviation) and higher values of wave propagation velocities/wavelengths were noticed during ASOND in contrast to MAMJJ. The joint effect of higher values of mean background current velocities along the ISW traveling direction and a deeper pycnocline but less stratified (smaller N2, see Figure 12) may explain the increase of the propagation velocities/wavelength of the waves during

the boreal summer/fall in agreement with Barbot et al. (2021); Tchilibou et al. (2022). Barbot et al. (2021) found that a deeper pycnocline due to large anticyclonic eddies of the NBC mostly from August to November increases the horizontal surface wavelengths of both mode-1 and mode-2 IT with a stronger impact on mode-1. No seasonal changes of mode-2 propagation velocities/wavelengths is found in contrast to the mode-1 ones. However, it is important to point out that the limited number of mode-2 wave samples in area A during MAMJJ likely impaired our analysis. Using the TGE, the seasonal variability of

predicted mode-1 and mode-2 waves was examined. Although the differences in the mode-1 propagation velocities/wavelength between ASOND and MAMJJ are underestimated by the theoretical method, TGE could reproduce the seasonal differences of the IT wavelength giving us confidence in our previous results and supporting our analysis principally considering our unbalanced data set according to the seasons. Furthermore, the TGE could reproduce the higher diversity of ISWs in terms of wavelength during the period of ASOND in agreement with our satellite measurements.

Finally, the comprehensive data set constructed during our analysis will support further studies to assess the impact of ISWs on the biological/biogeochemical dynamics in the study area with an emphasis on their impact on the phytoplankton biomass spatio-temporal variability.

### *Impact of NBC*

The circulation can likely be pointed to as one important factor in the change of the velocities of the waves according

to the different propagation direction paths. The eastward reinforcement of the NECC during ASOND seems to play a role in refracting the waves northeast, increasing their velocities with increasing east traveling direction component and giving them an extra offshore acceleration. Magalhães et al. (2016) found an increase of 30% in the ISWs velocities during ASOND based on the study of two showcases (one of them from May and the other one from October). The authors associated the seasonal differences in the propagation velocities/wavelengths with the variability of the NECC, which refracts the wave and

provides an additional (positive) component along the ISW traveling direction. The impact of the circulation in the propagation velocities/wavelength is more evident for mode-2 waves. According to Rainville and Pinkel (2006), the mesoscale ocean circulation changes the ISW propagation path, and group and phase velocities of all wave modes, however, the impact increases with the increasing of the mode numbers. Furthermore, the presence of waves with higher diversity in terms of their propagation velocities propagating in a wider pathway (see Figure 14) during ASOND seems to be connected to the intensification of the

currents and mesoscale activity in this season, which brings a higher variability in the shear/circulation conditions. According



to Tchilibou et al. (2022), during ASOND the background circulation induces refraction and branching of the IT direction propagation beams in the study area. The more dynamic circulation during ASOND seems also to laterally spread the waves in the study area, extending their penetration further North.

### *Aliasing effect*

It is important to point out that we cannot rule out the sampling restrictions due to the aliasing effect connected to the sun-synchronous satellite orbit which can result in images acquired during a similar flood–ebb phase of the semi-diurnal tide (da Silva et al., 2015) and to the location of the sun glint areas. Further analysis can be performed focusing on the construction in the study area of a more balanced data set according to both the different seasons and wave baroclinic modes by considering the use of different optical sensors/satellites such as AQUA/MODIS, MSI/Sentinel-2, and OLCI/Sentinel-3.

*Author contributions.* The remote sensing data processing was made by C.R.d.M., C.A.D.L., and M.C.B.R. with help of T.K.T. Tide simulation was made by A.K.L. Analysis was performed and discussed by C.R.d.M. with help of A.K.L., V.V., J.C.B.d.S., J.M.M., and C.A.D.L. The manuscript was written with help of all authors.

*Competing interests.* The authors declare that they have no known competing financial interests or personal relationships that could have appeared to influence the work reported in this paper.

*Financial support.* This work this supported by CNES funding in the frame of the MIAMAZ TOSCA project. C.A.D.L. is funded by the Ministry of Science, Technology, and Innovation and the Brazilian Navy (CNPQ/MCTI 06/2020 - grant #440852/2020−0), and the research project AtlantECO (H2020 BG-08-2018-2019, grant agreement N°862923). M.C.B.R. is funded by the Coordenação de Aperfeiçoamento de Pessoal de Nível Superior – Brasil (CAPES) – Finance Code 001. J.C.B.d.S. is funded by the Portuguese funding agency Fundação para a Ciência e Tecnologia (FCT) under project IDB/04683/2020. J.M.M. is supported by FCT – Portuguese Foundation for Science and 440  Technology under contracts UIDB/04423/2020 and UIDP/04423/2020.

*Acknowledgements.* The authors would like to thank the NASA's Earth Science Data System, ESDS for providing the MODIS/TERRA data, the Mercator Ocean International as part of the Copernicus Programme for providing the The Global Ocean Ensemble Physics Reanalysis (EPR) data. This work is a contribution to the project "Amazomix"




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
