# Peer review of "Spatial and temporal variability of mode-1 and mode-2 internal solitary waves from MODIS/TERRA sun glint off the Amazon shelf."

_EGUsphere, 2022_

## Referee Comment (RC1)

*Review of*

**"Spatial and temporal variability of mode-1 and mode-2 internal solitary waves from MODIS/TERRA sun glint off the Amazon Shelf"**

*by C. R. de Macedo, A. Koch-Larrouy, J. C. B. da Silva, J. M. Magalhães,*
*C. A. D. Lentini, T. K. Tran, M. C. B. Roxa and V. Vantrepotte*

This paper documents the spatial and temporal variability of mode-1 and mode-2 internal solitary waves off the Amazon shelf using sun glint images from MODIS/TERRA satellites. Only off-shore propagating waves are considered. The images span a period of 17 years and two time periods are considered because of differences in the background stratification and hence wave properties: spring (March to June) and fall (August to December). The most interesting result is the number of presumed mode-2 ISWs there are and their seasonal variation.

The paper makes a valuable contribution however I did find it rather dry and tedious to read. Perhaps some of the details can be put in appendices or supporting material? With that in mind the authors could try to reduce the paper by a few pages and think about what is really essential. The paper also suffers from a lack of physical insight - there is a lot of reporting about what the images show and there could be more on why. There is some mention of the seasonal variation of the stratification and ocean currents. It would help if there were some plots to illustrate this and perhaps some plots showing the model structure in the different seasons and in different areas. How are the variations in the number or type of ISW related to variations in internal tide energy flux?

The writing could also be improved. I think it needs a rather thorough revision before I could recommend it for publication.

**Comments**

1. ISWs are categorized as mode-1 or mode-2 waves based on the distance between successive wave packets: if the separation distance is approximately equal to a mode-1/mode-2 internal tide wavelength they are called mode-1/mode-2 ISWs. While this assumption seems a reasonable one there is no in situ data reported on that confirms the vertical structure of the ISWs. Can the authors comment on this?

2. In section 2.3 the viscous Taylor-Goldstein Equation (TGE) is discussed. It is used to compute the mode-1 and mode-2 internal tide wavelengths and propagation speeds. Rotational effects are not included which is reasonable for such a low latitude but perhaps should be commented on. The values of the eddy viscosity and diffusivity used in the calculations are not provided. What values were used and why? How much does the inclusion of these terms affect the results?

3. At times the discussion of the regions where the waves appear is confusing. For example in the first paragraph of section 3 on page 8 sites A, B, F, etc. are referred to. These appear to refer to regions where ISWs appear. Later the

terminology areas A, B, etc. is used (see top of page 15 for example). This change in terminology is confusing. In Figure 2 there are locations labelled A, B, C and D. These I take to be generation sites. But in the text sites A, B, etc. are often large areas in which ISW surface signatures are seen. And the various sites appear to overlap. For example it appears that site A is not disjoint from site D. It would be helpful to add boxes to figure 2 showing where site (or area) A, B, etc. are. It would also be helpful to use consistent terminology that clearly distinguishes the generate site and the region where waves generated at the site are observed. Please also explain how the generation site of an ISW is determined. For example in Figure 5 it is stated that the red lines indicate ISW emanating from generation site F. How do you know they came from F and not E?

Also, the text refers to site F in the first paragraph of section 3 (page 8) when Figure 2 is being discussed but there is no E or F on Figure 2. One has to wait for Figure 5 to see where they are.

4. Is there some meaning to the ordering of the generation sites A, B, C, etc.? They are not east to west for example. Are they ordered in order of importance in generating ISWs? IT energy flux amplitudes?

5. Page 9, line 201. How were the number of mode-1 and mode-2 wave signatures normalized?

6. Figure 8 and 9. It appears the captions are mixed up: the caption for figure 8 should be for figure 9?

7. Lines 296–297. Here it is stated that the TGE calculations do not reproduce difference in propagation speeds at spring and neap tides. I don't understand why this is stated as the TGE equation is a linear equation which can't take into account changes in the strength of the barotropic tidal currents.

8. Line 140. "The buoyancy perturbations vertical and velocity ..." doesn't make sense.

9. Line 160. "... the phase speed is depicted complex number whose imaginary part ..." is ungrammatical. It doesn't make sense anyway because in (9) $c$ is defined to be $\omega/k$ where $\omega$ is the negative of the imaginary part of $\sigma$ so $c$ is real.

10. Line 209. When I look at figure 4 it doesn't look like the probability of finding a mode- signature is more than 4 times the probability of finding a mode-2 signature. Please explain.

11.

12.

13.

14.

15.

16.

17.

18.

19.

20.

21.

22.

23.

24.

25.

26.

27.

28.

29.

30.

31.

---

## Referee Comment (RC2)

**Review of "Spatial and temporal variability of mode-1 and mode-2 internal**

**solitary waves from MODIS/TERRA sun glint off the Amazon shelf."**

**By Carina Regina de Macedo et al**

The paper is focused on the characterization of ISWs off the amazon shelf from MODIS satellite imagery. The most important result is the characterization of recurrent mode 2 ISWs and the impact of seasonality/circulation on wavelength and phase speed propagation directions. Regarding mode-1 ISWs most of the results confirm the analysis of Marghales et al (2016) based on SAR data; more interesting is the detection of mode 2 ISWs which are more impacted by seasonality and circulation than mode-1 waves.

I believe the paper could be accepted for publication after moderate/minor revisions

In general the analysis is convincing. I think yet that the background conditions obtained from the model for the circulation and IT generation could be added to further evidence the impact of circulation on the figures. I also think that a ray tracing computation for mode-1 and mode-2 following Rainville 2006 would be a nice addition to further illustrate the impact of refraction by the circulation.

Another point is that I do not really agree with the fact that the authors rule out KdV arguing it does only apply for flat bottom, it is also the case for their modal decomposition! I believe some KdV estimate of the phase speed would probably mostly fill the gap between the phase speed they observe from satellite and the linear modal phase speed. The problem is more than they can't estimate the amplitude of the ISWs to get the Kdv phase speed. All in all despite the systematic underestimation of phase speed/wavelength by the linear model, the difference of characteristics between mode-1 and mode-2 and the bimodal distribution of ISWs characteristics is enough to convince the reader that both ISW amode-1 and mode-2 are observed.

IN the abstract and the conclusion, the authors suggest at generation of ISW at the point of IT beam reflection, I guess this corresponds to the local generation mechanism described by Gerkema, if so he should be cited and this mechanism more thoroughly discussed.

The abstract refers to very specific labels/features of the figure and is mostly impossible to understand without seeing the figure. Although it is fine to put some quantitative results in the abstract I would suggest to delete too much specific reference to fig labels etc…

I have several more specific comments/questions below.

Abstract L14-15 seems contradictory with previous sentence

L61 shift what? the depth of max amplitude of displacement i-e velocity node?

L146-147 how these diffusivities and viscosities are estimated, what is their impact on the phase speed?

L187 On figure 2 A and B are between isobath 500 and 2000 m not 200 m

L194-195 cloud cover could definitely be as strong biased isn't it possible to get the distribution for clear sky images, or even simply the mean crest length and stdv for clear sky images?

L202 why normalized ?

L203-205 It seems really difficult to conclude anything on the seasonality r of the number of signatures, if we restrict to months with at least 5-10 images I don't see any significant differences.

L209 I don't get this sentence, you find mode 1 every year according to Fig4 b and mode 2 every year except 2009, what do you mean by "the annual probability of finding a mode-1 signature is more than 4 times the probability of finding a mode-2 one"

Paragraph starting at L219 seems to me to describe the method to separate mode 1 and mode 2 wave and should therefore be introduced before Fig.5

L284 how do you define near spring?  +/- 7 days after peak spring?

L296-297 difference of phase speed between spring and neap tide is typically a result of amplitude/nonlinearity so it can't be reproduced

L363 I think the authors describe the generation of ISW at the point of IT beam reflection, I guess this corresponds to the local generation mechanism described by Gerkema, if so he should be cited I thonk.

---

## Author Comment (AC1)

**Spatial and temporal variability of mode-1 and mode-2 internal solitary waves from MODIS/TERRA sun glint off the Amazon shelf.**

Carina Regina de Macedo[1,2], Ariane Koch-Larrouy[2], Jose Carlos Bastos da Silva[3,4], Jorge Manuel Magalhães[3,5], Carlos Alessandre Domingos Lentini[6,7,8], Trung Kien Tran[1], Marcelo Caetano Barreto Rosa[7], Vincent Vantrepotte[1]

[1]Univ. Lille, CNRS, Univ. Littoral Côte d'Opale, IRD, UMR 8187 - LOG - Laboratoire d'Océanologie et de Géosciences, F-59000Lille, France.

[2]LEGOS, Université de Toulouse, CNES, CNRS, IRD, UPS, Toulouse, France.

[3]Department of Geosciences, Environment and Spatial Planning, Faculdade de Ciências da Universidade do Porto, Rua do Campo Alegre 687, 4169-007, Porto, Portugal.

[4]Instituto de Ciências da Terra, Polo Porto, Universidade do Porto, Rua do Campo Alegre 687, 4169-007, Porto, Portugal.

[5]CIIMAR, Universidade do Porto, Rua dos Bragas 289, 4050-123, Porto, Portugal.

[6]Department of Earth and Environment Physics, Physics Institute, Ondina Campus, Federal University of Bahia—UFBA, Salvador, Bahia, Brazil.

[7]Department of Oceanography, Geosciences Institute, Campus Ondina, Federal University of Bahia —UFBA, Salvador, Bahia, Brazil.

[8]Interdisciplinary Center for Energy and Environment (CIEnAm), Federal University of Bahia UFBA, Salvador, Bahia, Brazil.

**We thank the reviewer for taking the time to review our manuscript and especially for the valuable comments regarding the impact of the background current in the mode-2 waves according to the seasons, the use of the kdv, and the valuable new reference. In the following, our responses to the reviewer's comments are in bold black colors.**

The paper is focused on the characterization of ISWs off the Amazon shelf from MODIS satellite imagery. The most important result is the characterization of recurrent mode 2 ISWs and the impact of seasonality/circulation on wavelength and phase speed propagation directions. Regarding mode-1 ISWs most of the results confirm the analysis of Margalhães et al (2016) based on SAR data; more interesting is the detection of mode-2 ISWs which are more impacted by seasonality and circulation than mode-1 waves.

I believe the paper could be accepted for publication after moderate/minor revisions.

In general, the analysis is convincing. I think yet that the background conditions obtained from the model for the circulation and IT generation could be added to further evidence the impact of circulation on the figures. I also think that a ray tracing computation for mode-1 and

mode-2 following Rainville 2006 would be a nice addition to further illustrate the impact of refraction by the circulation.

**ANSWER:**

**New calculations of the time-space variability of the waves' phase velocity in the study area were used as a proxy for change in the wave's propagation velocity (refraction). This methodology was chosen because seemed to be the more direct way to take into account the effects of both background circulation and stratification through the TGE. In fact, results show higher variability of phase velocity for mode-2 waves than mode-1, showing mode-2 waves as more sensitive to the background condition, as suggested by our measurements and the findings in Rainville and Pinkel (2006). Furthermore, in area 2, higher variability aligned to the NECC was found during ASOND, in good agreement with our results. Indeed, the authors agree that in the future the ray tracing computations for mode-1 and mode-2 ISWs would be a great addition for further understanding the waves in our study region, especially considering the very different background current conditions according to the seasons. For more details, please, see pages 18-19, lines 286-294, and Figure 15.**

Another point is that I do not really agree with the fact that the authors rule out KdV arguing it does only apply for flat bottom, it is also the case for their modal decomposition! I believe some KdV estimate of the phase speed would probably mostly fill the gap between the phase speed they observe from satellite and the linear modal phase speed. The problem is more than they can't estimate the amplitude of the ISWs to get the Kdv phase speed. All in all despite the systematic underestimation of phase speed/wavelength by the linear model, the difference of characteristics between mode-1 and mode-2 and the bimodal distribution of ISWs characteristics is enough to convince the reader that both ISW mode-1 and mode-2 are observed.

**ANSWER**

**Considering that, in the study area, the upper layer thickness ($h_1$) is smaller than the lower layer ($h_2$) (see Figure 1, where a representative profile of density in the study area is shown based on the climatology EPR data), the wave interface displacement will be a depression. So, the nonlinear phase speed ($c$), following the KdV equation, can be calculated as follow (Jeans, 1995):**

$$c = c_0 \left( 1 - \frac{\alpha \eta_0}{3 c_0} \right)$$

where $\eta_0$ being the maximum wave elevation, and $\alpha$ being the nonlinear coefficient:

$$\alpha = -\frac{3c_0\left(1+r\right)}{2h_1}$$

and $r = h_1/h_2$.

This means that a nonlinear component directly proportional to the wave elevation should be added to the linear phase speed of the waves. Taking from Figure 2, $h_1 \sim 100$ m and $h_1 \sim 3800$ m, and considering $\eta_0 = 100$ m (Brandt, 2002), and $c_0 = 2.16$ $m.s^{-1}$ for mode-1 waves (considering a two-layer model Gill, A. E. (1982). *Atmosphere-ocean dynamics* (Vol. 30). Academic press.), the nonlinear component is about 1.26 $m.s^{-1}$, i.e., $c = 3.42$ $m.s^{-1}$. This value is about 16% overestimated compared to our remote sensing measurements (mean mode-1 velocity in area 2 of 2.94 $m.s^{-1}$) and also to values presented by Magalhães et al. 2016 (who find a mean mode-1 velocity of 3.1 $m.s^{-1}$). This information was added to the text, please, see page 25, lines 439-440.

[Figure]

**Figure 1 - Representative profile of Density in Amazon shelf from climatology EPR data.**

In the abstract and the conclusion, the authors suggest at generation of ISW at the point of IT beam reflection, I guess this corresponds to the local generation mechanism described by Gerkema, if so he should be cited and this mechanism more thoroughly discussed.

**ANSWER:**

**The bibliography "Gerkema, T. (2001). Internal and interfacial tides: beam scattering and local generation of solitary waves. Journal of Marine Research, 59(2), 227-255"**

**was added to the manuscript. The mechanism was thoroughly discussed based on that bibliography (please, see Page 20, lines 305-307).**

The abstract refers to very specific labels/features of the figure and is mostly impossible to understand without seeing the figure. Although it is fine to put some quantitative results in the abstract I would suggest to delete too much specific reference to fig labels etc…

**ANSWER:**
**Specific labels/features were removed from the abstract as suggested by the reviewer.**

I have several more specific comments/questions below.

1. Abstract L14-15 seems contradictory with previous sentence.

**ANSWER:**
**We have changed the text to be more comprehensive: "The mean mode-1 and mode-2 propagation velocities/wavelengths do not show significant differences according to their IT generation sites. A larger proportion of mode-2 waves is likely linked to shallower pycnocline with higher maximum values" (Page 1, lines 10-12).**

2. L61 shift what? the depth of max amplitude of displacement i-e velocity node?

**ANSWER:**
**Deeper pycnocline shift the first extrema of the vertical modal structure to deeper water layers and the second one to shallower layers. This information is added in the text as follows: "Analyzing the vertical modal structure for mode-2 IT, a deeper pycnocline seems to shift the extrema of the modes toward intermediate water layers (i.e., the first extremum is deeper and the second one is shallower" (see Page 3, lines 56-57).**

3. L146-147 how these diffusivities and viscosities are estimated, what is their impact on the phase speed?

**ANSWER:**
**The original T-G equation was solved assuming inviscid flow. Lian's numerical method, which was used in our paper, is indeed for the viscous Taylor-Goldstein equation. The method includes the effects of viscosity and diffusivity, which in origin**

we choose to be the kinematic viscosity of water at 20 °C, i.e., 1 cSt or $10^{-6} \ m/s^2$ and the thermal diffusivity of water at 20 °C, i.e., 1.43$.10^{-7} \ m/s^2$ (this information is added in the manuscript, please, see page 25, lines 430-434). We have done tests and for internal tidal waves (of various modes) it does not make any difference to use one or another.

4. L187 On figure 2 A and B are between isobath 500 and 2000 m not 200 m.

**ANSWER:**

**Indeed, the Isobath's legend was messed up. We corrected it in the Figures and in the text (please, see Page 7, lines 156-157).**

5. L194-195 cloud cover could definitely be as strong biased isn't it possible to get the distribution for clear sky images, or even simply the mean crest length and stdv for clear sky images?

**ANSWER:**

**Indeed, cloud cover image distribution over a year may induce some bias on the crest length statistics, underestimating its actual arc length. Working with optical/passive imagery systems such as MODIS is the price to pay. On the other hand, MODIS-Terra statistics can be complementary to the SAR-derived one, as its spatial and temporal coverage are usually superior. Moreover, MODIS-derived data allows extracting other essential statistics such as the number of waves in a packet, inter and intra-packet distances.**

6. L202 why normalized?

**ANSWER:**

**More signatures during the months of ASOND, for example, could be only a reflection of the higher number of images with less cloudy coverage. So, we have done the normalization (number of detected signatures according to the different wave modes divided by the total number of images containing at least one clear signature) to attenuate the impact of cloud coverage seasonability.**

7. L203-205 It seems really difficult to conclude anything on the seasonality of the number of signatures, if we restrict to months with at least 5-10 images I don't see any significant differences.

**ANSWER**

**The authors agree with the reviewer and we have changed this in the manuscript: "Because of the lack of acquisitions for some months, no evident seasonal variability is found" (please, see pages 7-8, lines 172-173).**

8. L209 I don't get this sentence, you find mode 1 every year according to Fig4 b and mode 2 every year except 2009, what do you mean by "the annual probability of finding a mode-1 signature is more than 4 times the probability of finding a mode-2 one".

**ANSWER:**

**The analysis was redone and the phrase was changed to: "The total number of detected mode-1 waves is about 3 times the number of mode-2 ones" (please, see Figure 4, and page 8, lines 174-175).**

9. Paragraph starting at L219 seems to me to describe the method to separate mode 1 and mode 2 wave and should therefore be introduced before Fig.5.

**ANSWER:**

**This paragraph was shifted before Figure 5, as suggested by the reviewer.**

10. L284 how do you define near spring? +/- 7 days after peak spring?

**ANSWER:**

**Yes, near spring tide is defined as +/- 7 days after spring tide peak. This information was added in the manuscript: "Near spring tide is defined as +/- 7 days after spring tide peak" (please, see Page 13, line 230).**

11. L296-297 difference of phase speed between spring and neap tide is typically a result of amplitude/nonlinearity so it can't be reproduced.

**ANSWER:**

**The authors agree with the reviewer's comment and this part was removed from the manuscript.**

12. L363 I think the authors describe the generation of ISW at the point of IT beam reflection, I guess this corresponds to the local generation mechanism described by Gerkema, if so he should be cited I think.

**ANSWER:**

**The bibliography "Gerkema, T. (2001). Internal and interfacial tides: beam scattering and local generation of solitary waves. Journal of Marine Research, 59(2), 227-255" was added to the manuscript (please, see Page 20, lines 305-307).**

---

## Author Comment (AC2)

**Spatial and temporal variability of mode-1 and mode-2 internal solitary waves from MODIS/TERRA sun glint off the Amazon shelf.**

Carina Regina de Macedo[1,2], Ariane Koch-Larrouy[2], Jose Carlos Bastos da Silva[3,4], Jorge Manuel Magalhães[3,5], Carlos Alessandre Domingos Lentini[6,7,8], Trung Kien Tran[1], Marcelo Caetano Barreto Rosa[7], Vincent Vantrepotte[1]

[1]Univ. Lille, CNRS, Univ. Littoral Côte d'Opale, IRD, UMR 8187 - LOG - Laboratoire d'Océanologie et de Géosciences, F-59000Lille, France.
[2]LEGOS, Université de Toulouse, CNES, CNRS, IRD, UPS, Toulouse, France.
[3]Department of Geosciences, Environment and Spatial Planning, Faculdade de Ciências da Universidade do Porto, Rua do Campo Alegre 687, 4169-007, Porto, Portugal.
[4]Instituto de Ciências da Terra, Polo Porto, Universidade do Porto, Rua do Campo Alegre 687, 4169-007, Porto, Portugal.
[5]CIIMAR, Universidade do Porto, Rua dos Bragas 289, 4050-123, Porto, Portugal.
[6]Department of Earth and Environment Physics, Physics Institute, Ondina Campus, Federal University of Bahia—UFBA, Salvador, Bahia, Brazil.
[7]Department of Oceanography, Geosciences Institute, Campus Ondina, Federal University of Bahia —UFBA, Salvador, Bahia, Brazil.
[8]Interdisciplinary Center for Energy and Environment (CIEnAm), Federal University of Bahia UFBA, Salvador, Bahia, Brazil.

**We thank the reviewer for taking the time to review our manuscript and especially for the valuable comments regarding the manuscript structure, the Taylor-Goldstein Equation, and the reinforcement of the manuscript's physical approach. In the following, our responses to the reviewer's comments are in bold black colors.**

This paper documents the spatial and temporal variability of mode-1 and mode-2 internal solitary waves off the Amazon shelf using sun glint images from MODIS/TERRA satellites. Only off-shore propagating waves are considered. The images span a period of 17 years and two time periods are considered because of differences in the background stratification and hence wave properties: spring (March to June) and fall (August to December). The most interesting result is the number of presumed mode-2 ISWs there are and their seasonal variation.

The paper makes a valuable contribution however I did find it rather dry and tedious to read. Perhaps some of the details can be put in appendices or supporting material? With that in

mind the authors could try to reduce the paper by a few pages and think about what is really essential.

**ANSWER:**
**The paper was shortened and some details were added as supporting material (please, see Appendix A).**

The paper also suffers from a lack of physical insight - there is a lot of reporting about what the images show and there could be more on why. There is some mention of the seasonal variation of the stratification and ocean currents. It would help if there were some plots to illustrate this and perhaps some plots showing the model structure in the different seasons and in different areas.

**ANSWER:**
**In our manuscript, we have already included discussions of stratification and current according to the different areas and seasons. The stratification in the different areas was discussed based on the mean and standard variation of the Brunt–Väisälä frequency calculated from daily Global Ocean Ensemble Physics Reanalysis (EPR) data. A slightly shallower pycnocline with higher maximum values of the Brunt–Väisälä frequency in area 3 was suggested as one contribution for the higher proportion of mode-2 ISWs in that area (see Figure 6-(a)). The seasonal variability in stratification was also discussed and pointed out as one factor to explain the increase of the ISW velocities during ASOND (see Figure 12-(a)). More detailed information about that can be found in Tchilibou et al. (2022). Furthermore, we included an analysis of the current (taken from daily EPR data) based on the decomposition of the current velocities in the ISW traveling direction. So, we could have evidence of the impact of the NECC during ASOND (Figure 12-(b)) in area 2. The mean current during ASOND and MAMJJ are also discussed (Figure 14) and, during ASOND, the reinforcement of the NECC and higher mesoscale activity are pointed out as one factor that spread the waves in the study area. To have a more physical approach, as suggested by the reviewer, the time-space variability of the wave's phase velocity associated with changes in the background and stratification was exploited as a proxy of the variability of the wave's propagation direction (refraction). Results showed higher variability for mode-2 phase velocity supporting our results which show mode-2 waves as more impacted by the background current. Furthermore, area 2 showed a higher variability of the phase velocity aligned to the NECC during ASOND. For more details, please see page 18, lines 286-294, and Figure 15.**

How are the variations in the number or type of ISW related to variations in internal tide energy flux?

**ANSWER:**

**The variation in the type of ISWs is discussed based on the results of internal tide energy flux presented in Tchilibou et al. (2022): "According to Tchilibou et al. (2022), during MAMJJ the mode-1 and mode-2 baroclinic fluxes from IT generation point A (contained in area 2) propagate further north than during ASOND. The stronger circulation and mesoscale activity during the latter season are pointed as factors that largely block the energy flux at 6°N. The IW signatures mapped from RS data do not reproduce that behavior as it may be due to our sample restrictions during MAMJJ. During ASOND, the baroclinic flux is eastward deviated by the background circulation east of 45°W (Tchilibou et al., 2022). This behavior is reproduced by the IW signatures which seem to be laterally spread in the study area (refracted northeast) by the reinforcement of the NECC (see Figure 13-(b)). Furthermore, during ASOND the flux coming from the IT site D (contained in area 2) divides into two, creating a northwest branching. In Figure 13-(b) the branching is visible in the ISW satellite measurements. The mode-2 baroclinic fluxes coming from D and A have a more dissociate path (Tchilibou et al., 2022) compared to mode-1. In Figure 6, the identification of mode-2 ISW signatures coming from D is more evident than for mode-1." (please, see pages 21-22, lines 368-377 ).**

The writing could also be improved. I think it needs a rather thorough revision before I could recommend it for publication.

**ANSWER**

**We corrected the English grammar in the text, making it more comprehensive and easy to read.**

Comments

1. ISWs are categorized as mode-1 or mode-2 waves based on the distance between successive wave packets: if the separation distance is approximately equal to a mode-1/mode-2 internal tide wavelength they are called mode-1/mode-2 ISWs. While this assumption seems a reasonable one there is no in situ data reported on that confirms the vertical structure of the ISWs. Can the authors comment on this?

**ANSWER:**

**The mode-2 waves described in our paper relate to large-scale mode-2-like waves with scale (wavelength) associated with mode-2 internal tides. That waves have a wavelength of the order of kilometers being different from the classic mode-2 ISWs with a wavelength of the order of hundred meters. The coexistence of mode-2 solitary-like waves and mode-1 ISW has been documented in the literature, for example, on the Mascarene Ridge (da Silva et al. 2015) and on the Andaman Sea (Magalhaes et al., 2020, Magalhaes and da Silva, 2018). The mode-2 internal tide was discussed in Tchilibou et al., 2022 and Barbot et al., 2021 with the mode-2 wavelength range inferred from the circulation model being in good agreement with our results retrieved from remote sensing data. Unfortunately, no in situ data showing the vertical structure of mode-2-like waves in our study area is yet available (probably by the end of this year we will start getting some results from the AMAZOMIX oceanographic cruiser, for more info please see https://campagnes.flotteoceanographique.fr/campagnes/18001364/ ). This discussion was inserted in the manuscript. (please, see page 4, lines 92-94, and page 20, lines 312-314).**

2. In section 2.3 the viscous Taylor-Goldstein Equation (TGE) is discussed. It is used to compute the mode-1 and mode-2 internal tide wavelengths and propagation speeds. Rotational effects are not included which is reasonable for such a low latitude but perhaps should be commented on. The values of the eddy viscosity and diffusivity used in the calculations are not provided. What values were used and why? How much does the inclusion of these terms affect the results?

**ANSWER**

**Lian's method for solving the viscous TGE does not take into account the Coriolis force due to the Earth's rotation. This information was added to our manuscript and commented on according to our study area (please, see Appendix A, page 25, lines 429-430). The original T-G equation was solved assuming inviscid flow (Miles, J. W. (1961). On the stability of heterogeneous shear flows.** *Journal of Fluid Mechanics*, *10*(4), 496-508). **Lian's numerical method, which was used in our paper, is indeed for the viscous Taylor-Goldstein equation. The method includes the effects of viscosity and diffusivity, which in origin we choose to be the kinematic viscosity of water at 20 °C, i.e., 1 cSt or $10^{-6}$ $m/s^2$ and the thermal diffusivity of water at 20 °C, i.e., 1.43.$10^{-7}$ $m/s^2$ (this information is added in the manuscript). We have done tests and for**

**internal tidal waves (of various modes) it does not make any difference to use one or another. That discussion was added to the manuscript, please, see Page 25, lines 430-434.**

3. At times the discussion of the regions where the waves appear is confusing. For example in the first paragraph of section 3 on page 8 sites A, B, F, etc. are referred to. These appear to refer to regions where ISWs appear. Later the terminology areas A, B, etc. is used (see top of page 15 for example). This change in terminology is confusing.

In Figure 2 there are locations labelled A, B, C and D. These I take to be generation sites. But in the text sites A, B, etc. are often large areas in which ISW surface signatures are seen. And the various sites appear to overlap. For example it appears that site A is not disjoint from site D. It would be helpful to add boxes to figure 2 showing where site (or area) A, B, etc. are. It would also be helpful to use consistent terminology that clearly distinguishes the generate site and the region where waves generated at the site are observed.

Please also explain how the generation site of an ISW is determined. For example in Figure 5 it is stated that the red lines indicate ISW emanating from generation site F. How do you know they came from F and not E? Also, the text refers to site F in the first paragraph of section 3 (page 8) when Figure 2 is being discussed but there is no E or F on Figure 2. One has to wait for Figure 5 to see where they are.

**ANSWER:**
**We have changed the text to use consistent terminology to distinguish the IT generation sites (from A to F) from the areas/regions where ISWs signatures have been found (called hereafter areas 1, 2, and 3). Boxes showing areas 1, 2, and 3 were added in Figure 2 as well as the IT generation points E and F, as suggested. The likely generation sites of ISWs are determined by analysis of the M2 baroclinic flux described in Tchilibou et al., 2022. This information was added in the text. Regarding the ISWs in area 1, we can not be sure if the origin of the waves is the sites E or F, indeed. So, we have modified the text (please, see Pages 6-7, lines 147-157).**

4. Is there some meaning to the ordering of the generation sites A, B, C, etc.? They are not east to west for example. Are they ordered in order of importance in generating ISWs? IT energy flux amplitudes?

**ANSWER:**
**The sites are organized according to their energy flux amplitude (same order presented in Tchilibou et al. (2022)). The generation areas A and B were identified first**

**by Magalhães et al., 2016, who ordered them according to their energy flux amplitudes. Tchilibou et al. (2022) have identified more 4 areas (from C to F) which were organized as well according to their energy flux (still sites A and B had the highest energy flux). This info is added in the legend of Figure 2.**

5. Page 9, line 201. How were the number of mode-1 and mode-2 wave signatures normalized?

**ANSWER:**
**The signatures were normalized by computing the number of signatures per year/month divided by the number of images containing at least one clear signature. The information was added to the manuscript, see Page 7, lines 169-170.**

6. Figure 8 and 9. It appears the captions are mixed up: the caption for figure 8 should be for figure 9?

**ANSWER:**
**Indeed, the captions of Figures 8 and 9 were mixed up. We corrected it.**

7. Lines 296–297. Here it is stated that the TGE calculations do not reproduce difference in propagation speeds at spring and neap tides. I don't understand why this is stated as the TGE equation is a linear equation which can't take into account changes in the strength of the barotropic tidal currents.

**ANSWER:**
**The authors agree with the reviewer's comment and this part was removed from the manuscript.**

8. Line 140. "The buoyancy perturbations vertical and velocity ..." doesn't make sense.

**ANSWER:**
**We replaced this with "The vertical velocity perturbation and the buoyancy perturbation (i.e., $w'$ and $b'$, respectively)" (see page 24, line 409).**

9. Line 160. "... the phase speed is depicted complex number whose imaginary part ..." is ungrammatical. It doesn't make sense anyway because in (9) c is defined to be $\omega/k$ where $\omega$ is the negative of the imaginary part of $\sigma$ so c is real.

**ANSWER:**

**This information was removed from the text, as suggested by the reviewer.**

10. Line 209. When I look at figure 4 it doesn't look like the probability of finding a mode-signature is more than 4 times the probability of finding a mode-2 signature. Please explain.

**ANSWER:**

**Yes, indeed had a mistake in the analysis. It was corrected. Please, see the new Figure 4 and the comment on page 8, lines 174-175.**
* * *
**SECOND REVIEWER**

**We thank the reviewer for taking the time to review our manuscript and specially for the valuable comments regarding the impact of the background current in the mode-2 waves according to the seasons, the use of the kdv, and the valuable new reference. In the following, our responses to the reviewer's comments are in bold black colors.**

The paper is focused on the characterization of ISWs off the Amazon shelf from MODIS satellite imagery. The most important result is the characterization of recurrent mode 2 ISWs and the impact of seasonality/circulation on wavelength and phase speed propagation directions. Regarding mode-1 ISWs most of the results confirm the analysis of Margalhães et al (2016) based on SAR data; more interesting is the detection of mode-2 ISWs which are more impacted by seasonality and circulation than mode-1 waves.
I believe the paper could be accepted for publication after moderate/minor revisions.
In general, the analysis is convincing. I think yet that the background conditions obtained from the model for the circulation and IT generation could be added to further evidence the impact of circulation on the figures. I also think that a ray tracing computation for mode-1 and mode-2 following Rainville 2006 would be a nice addition to further illustrate the impact of refraction by the circulation.

**ANSWER:**

**New calculations of the time-space variability of the waves' phase velocity in the study area were used as a proxy for change in the wave's propagation velocity (refraction). This methodology was chosen because seemed to be the more direct way to take into account the effects of both background circulation and stratification through the TGE. In fact, results show higher variability of phase velocity for mode-2 waves than mode-1, showing mode-2 waves as more sensitive to the background condition, as suggested by our measurements and the findings in Rainville and Pinkel (2006). Furthermore, in area 2, higher variability aligned to the NECC was found during ASOND, in good agreement with our results. Indeed, the authors agree that in the future the ray tracing computations for mode-1 and mode-2 ISWs would be a great addition for further understanding the waves in our study region, especially considering the very different background current conditions according to the seasons. For more details, please, see pages 18-19, lines 286-294, and Figure 15.**

Another point is that I do not really agree with the fact that the authors rule out KdV arguing it does only apply for flat bottom, it is also the case for their modal decomposition! I believe some KdV estimate of the phase speed would probably mostly fill the gap between the phase speed they observe from satellite and the linear modal phase speed. The problem is more than they can't estimate the amplitude of the ISWs to get the Kdv phase speed. All in all despite the systematic underestimation of phase speed/wavelength by the linear model, the difference of characteristics between mode-1 and mode-2 and the bimodal distribution of ISWs characteristics is enough to convince the reader that both ISW mode-1 and mode-2 are observed.

**ANSWER**

**Considering that, in the study area, the upper layer thickness ($h_1$) is smaller than the lower layer ($h_2$) (see Figure 1, where a representative profile of density in the study area is shown based on the climatology EPR data), the wave interface displacement will be a depression. So, the nonlinear phase speed ($c$), following the KdV equation, can be calculated as follow (Jeans, 1995):**

$$c = c_0 \left( 1 - \frac{\alpha \eta_0}{3 c_0} \right)$$

**where $\eta_0$ being the maximum wave elevation, and $\alpha$ being the nonlinear coefficient:**

$$\alpha = -\frac{3 c_0 \left( 1 + r \right)}{2 h_1}$$

**and $r = h_1 / h_2$.**

This means that a nonlinear component directly proportional to the wave elevation should be added to the linear phase speed of the waves. Taking from Figure 2, $h_1 \sim 100$ **m and** $h_1 \sim 3800$ **m, and considering** $\eta_0 = 100$ **m (Brandt, 2002), and** $c_0 = 2.16$ $m.s^{-1}$ **for mode-1 waves (considering a two-layer model Gill, A. E. (1982).** *Atmosphere-ocean dynamics* **(Vol. 30). Academic press.), the nonlinear component is about 1.26** $m.s^{-1}$**, i.e.,** $c = 3.42$ $m.s^{-1}$**. This value is about 16% overestimated compared to our remote sensing measurements (mean mode-1 velocity in area 2 of 2.94** $m.s^{-1}$**) and also to values presented by Magalhães et al. 2016 (who find a mean mode-1 velocity of 3.1** $m.s^{-1}$**). This information was added to the text, please, see page 25, lines 439-440.**

[Figure]

**Figure 1 - Representative profile of Density in Amazon shelf from climatology EPR data.**

In the abstract and the conclusion, the authors suggest at generation of ISW at the point of IT beam reflection, I guess this corresponds to the local generation mechanism described by Gerkema, if so he should be cited and this mechanism more thoroughly discussed.

**ANSWER:**
**The bibliography "Gerkema, T. (2001). Internal and interfacial tides: beam scattering and local generation of solitary waves. Journal of Marine Research, 59(2), 227-255" was added to the manuscript. The mechanism was thoroughly discussed based on that bibliography (please, see Page 20, lines 305-307).**

The abstract refers to very specific labels/features of the figure and is mostly impossible to understand without seeing the figure. Although it is fine to put some quantitative results in the abstract I would suggest to delete too much specific reference to fig labels etc…

**ANSWER:**
**Specific labels/features were removed from the abstract as suggested by the reviewer.**

I have several more specific comments/questions below.

1. Abstract L14-15 seems contradictory with previous sentence.

**ANSWER:**
**We have changed the text to be more comprehensive: "The mean mode-1 and mode-2 propagation velocities/wavelengths do not show significant differences according to their IT generation sites. A larger proportion of mode-2 waves is likely linked to shallower pycnocline with higher maximum values" (Page 1, lines 10-12).**

2. L61 shift what? the depth of max amplitude of displacement i-e velocity node?

**ANSWER:**
**Deeper pycnocline shift the first extrema of the vertical modal structure to deeper water layers and the second one to shallower layers. This information is added in the text as follows: "Analyzing the vertical modal structure for mode-2 IT, a deeper pycnocline seems to shift the extrema of the modes toward intermediate water layers (i.e., the first extremum is deeper and the second one is shallower" (see Page 3, lines 56-57).**

3. L146-147 how these diffusivities and viscosities are estimated, what is their impact on the phase speed?

**ANSWER:**
**The original T-G equation was solved assuming inviscid flow. Lian's numerical method, which was used in our paper, is indeed for the viscous Taylor-Goldstein equation. The method includes the effects of viscosity and diffusivity, which in origin we choose to be the kinematic viscosity of water at 20 °C, i.e., 1 cSt or $10^{-6}\ m/s^2$ and the thermal diffusivity of water at 20 °C, i.e., $1.43.10^{-7}\ m/s^2$ (this information is added**

**in the manuscript, please, see page 25, lines 430-434). We have done tests and for internal tidal waves (of various modes) it does not make any difference to use one or another.**

4. L187 On figure 2 A and B are between isobath 500 and 2000 m not 200 m.

**ANSWER:**
**Indeed, the Isobath's legend was messed up. We corrected it in the Figures and in the text (please, see Page 7, lines 156-157).**

5. L194-195 cloud cover could definitely be as strong biased isn't it possible to get the distribution for clear sky images, or even simply the mean crest length and stdv for clear sky images?

**ANSWER:**
**Indeed, cloud cover image distribution over a year may induce some bias on the crest length statistics, underestimating its actual arc length. Working with optical/passive imagery systems such as MODIS is the price to pay. On the other hand, MODIS-Terra statistics can be complementary to the SAR-derived one, as its spatial and temporal coverage are usually superior. Moreover, MODIS-derived data allows extracting other essential statistics such as the number of waves in a packet, inter and intra-packet distances.**

6. L202 why normalized?

**ANSWER:**
**More signatures during the months of ASOND, for example, could be only a reflection of the higher number of images with less cloudy coverage. So, we have done the normalization (number of detected signatures according to the different wave modes divided by the total number of images containing at least one clear signature) to attenuate the impact of cloud coverage seasonability.**

7. L203-205 It seems really difficult to conclude anything on the seasonality of the number of signatures, if we restrict to months with at least 5-10 images I don't see any significant differences.

**ANSWER**

**The authors agree with the reviewer and we have changed this in the manuscript: "Because of the lack of acquisitions for some months, no evident seasonal variability is found" (please, see pages 7-8, lines 172-173).**

8. L209 I don't get this sentence, you find mode 1 every year according to Fig4 b and mode 2 every year except 2009, what do you mean by "the annual probability of finding a mode-1 signature is more than 4 times the probability of finding a mode-2 one".

**ANSWER:**
**The analysis was redone and the phrase was changed to: "The total number of detected mode-1 waves is about 3 times the number of mode-2 ones" (please, see Figure 4, and page 8, lines 174-175).**

9. Paragraph starting at L219 seems to me to describe the method to separate mode 1 and mode 2 wave and should therefore be introduced before Fig.5.

**ANSWER:**
**This paragraph was shifted before Figure 5, as suggested by the reviewer.**

10. L284 how do you define near spring? +/- 7 days after peak spring?

**ANSWER:**
**Yes, near spring tide is defined as +/- 7 days after spring tide peak. This information was added in the manuscript: "Near spring tide is defined as +/- 7 days after spring tide peak" (please, see Page 13, line 230).**

11. L296-297 difference of phase speed between spring and neap tide is typically a result of amplitude/nonlinearity so it can't be reproduced.

**ANSWER:**
**The authors agree with the reviewer's comment and this part was removed from the manuscript.**

12. L363 I think the authors describe the generation of ISW at the point of IT beam reflection, I guess this corresponds to the local generation mechanism described by Gerkema, if so he should be cited I think.

**ANSWER:**

The bibliography "Gerkema, T. (2001). Internal and interfacial tides: beam scattering and local generation of solitary waves. Journal of Marine Research, 59(2), 227-255" was added to the manuscript (please, see Page 20, lines 305-307).

---

## Author Comment (AC3)

**Spatial and temporal variability of mode-1 and mode-2 internal solitary waves from MODIS/TERRA sun glint off the Amazon shelf.**

Carina Regina de Macedo[1,2], Ariane Koch-Larrouy[2], Jose Carlos Bastos da Silva[3,4], Jorge Manuel Magalhães[3, 5], Carlos Alessandre Domingos Lentini[6,7,8], Trung Kien Tran[1], Marcelo Caetano Barreto Rosa[7], Vincent Vantrepotte[1]

[1]Univ. Lille, CNRS, Univ. Littoral Côte d'Opale, IRD, UMR 8187 - LOG - Laboratoire d'Océanologie et de Géosciences, F-59000Lille, France.
[2]LEGOS, Université de Toulouse, CNES, CNRS, IRD, UPS, Toulouse, France.
[3]Department of Geosciences, Environment and Spatial Planning, Faculdade de Ciências da Universidade do Porto, Rua do Campo Alegre 687, 4169-007, Porto, Portugal.
[4]Instituto de Ciências da Terra, Polo Porto, Universidade do Porto, Rua do Campo Alegre 687, 4169-007, Porto, Portugal.
[5]CIIMAR, Universidade do Porto, Rua dos Bragas 289, 4050-123, Porto, Portugal.
[6]Department of Earth and Environment Physics, Physics Institute, Ondina Campus, Federal University of Bahia—UFBA, Salvador, Bahia, Brazil.
[7]Department of Oceanography, Geosciences Institute, Campus Ondina, Federal University of Bahia —UFBA, Salvador, Bahia, Brazil.
[8]Interdisciplinary Center for Energy and Environment (CIEnAm), Federal University of Bahia UFBA, Salvador, Bahia, Brazil.

**We thank the reviewer for taking the time to review our manuscript and especially for the valuable comments regarding the ISW inter-packet distance variation according to the tidal cycle. In the following, our responses to the reviewer's comments are in bold black colors.**

The paper has improved significantly. The writing is still a bit cumbersome and difficult to read in places.

Comments

1. In the first sentence of section 2.1 it is stated that 140 images were used. I suggest replacing all statements that 'more than a hundred' images were used with '140 images' throughout the manuscript. 'More than 100' is not precise enough. More than 100 could mean 101, 10,000 or even more.

**ANSWER:**
**The total number of images was replaced throughout the whole manuscript, as suggested by the reviewer.**

2. The period MAMJJ includes spring and summer months but is referred to as 'spring' while ASOND includes summer and fall months and is referred to as summer/fall. Seems inconsistent not to refer to MAMJJ as spring/summer.

**ANSWER:**
**The authors agree with the reviewer and the terms boreal spring and boreal summer/fall were removed from the manuscript.**

3. Line 15. "play a role in refracting the waves towards the northeast'. Or 'bending the waves ...". I'd delete the 'which gives them an extra offshore acceleration'. The meaning of this is not very clear.

**ANSWER:**
**The "which gives them an extra offshore acceleration" was deleted, as suggested.**

4. Lines 29–30. Delete 'internal solitary waves' and just use the acronym as the acronym was defined on line 23.

**ANSWER:**
**"Internal solitary waves" was deleted.**

5. Line 33. Delete 'generate'. The hotspots aren't generated. Internal waves are generated.

**ANSWER:**
**The word "generated" was removed from the manuscript, as suggested.**

6. In the results section sub-patches are mentioned but not defined. Where in figure 2 are the subpatches? How were the distances between them, given in table 1, calculated?

**ANSWER:**
**The location of the sub-patches and their middle point used in the calculation of the distances were added in Figure 2-(b). More explanation is presented on page 7, lines 155-158.**

7. Mention is made of wavelengths and velocities estimated from the images. The wording often implies that the propagation velocity is estimated independently of the wavelength (e.g., "mean propagation velocity/wavelength varies ...' on line 238) but in reality the velocity is simply the distance between consecutive wave packets divided by the M2 tidal period. So if the wavelength increases by 10% then the velocity will as well. That the velocity is computed this way was not clear enough to me. It would help if the wording was changed to something like "... the wavelength increased by x% implying a corresponding increase in the propagation speed".
Also, I think using the term 'wavelength' instead of 'inter-packet distance' is not a great idea.

**ANSWER:**
**Taking into account the linearity of the ITs, the ISW inter-packet distances can be used as a proxy for the IT wavelengths (this information was added on page 5, lines**

**119-120). However, we agree with the reviewer, so the term wavelength when referring to ISWs was replaced by inter-packed distance in the whole manuscript, as suggested by the reviewer. Furthermore, as suggested, when referring to the propagation velocity, we add the term "corresponding velocity".**

8. Line 177. What does "calling for mode-1 waves" mean?

**ANSWER:**
**We changed to: "the group with higher inter-packet distance associated with mode-1 IT". Please, see page 8, line 183.**

9. Line 182. What is the simulated mode-1 mean propagation velocity?

**ANSWER:**
**We changed it to: "the calculated mode-1 mean propagation velocity is [...]". Please, see page 8, line 188.**

10. Lines 184–187. This seems a bit irrelevant and begs for an explanation of why the phase speed is proportional to the surface wave elevation and why it explains the higher underestimation of the mode-1 waves.

**ANSWER:**
**As suggested by the reviewer, this explanation was removed from the manuscript.**

11. Lines 197. Here it is stated that the mode-1 velocities are underestimated by 22%. What about the mode-2 velocities?

**ANSWER:**
**This information was added to the text: "The TGE allows a relevant prediction of the propagation velocity/wavelength distribution of mode-1 and mode-2 waves, with mode-1 velocities being underestimated by 22% and mode-2 by 3.7%" (page 11, line 205).**

12. Line 201. What is meant by 'joining area 2'? I do not see a green rectangle in Figure 7(b).

**ANSWER:**
**The green rectangle in Figure 7-(b) was missing. We corrected that.**

13. Line 227. Simplify: "... waves travel in a more eastward direction"

**ANSWER:**
**The sentence was changed as suggested by the reviewer.**

14. Line 238. Why are higher wavelengths associated with neap tides rather than spring tides where the stronger tides would suggest larger ITs being generated with larger propagation speeds.

**ANSWER:**

**We agree with the reviewer that ITs with higher velocities are expected during stronger tides, i.e., near spring tides. Indeed, our results show higher inter-packet distances (and, consequently, corresponding higher velocities) associated with the wave signatures found near neap tide. We argue that maybe this result could be associated with our unbalanced data set, which may hamper our analysis. This argumentation was inserted in the text, please, see page 14, lines 246-247.**

15. Line 344. Should 'Area A' be 'Area 2'?

**ANSWER:**

**Yes, "area A" is actually "area 2". We have corrected it.**

16. Lines 355–356. What is meant by 'refracting the waves northeast'. It seems like everything is propagating roughly northeast. Do you mean more eastward or more northward?

**ANSWER:**

**Indeed, we mean to say more eastward. This was changed in the manuscript according to the reviewer's comment.**

17. Lines 373–374. Mention is a northwest branch is made but I can't see anything propagating northwest in any of the Figures.

**ANSWER:**

**The authors agree with the reviewer. So, the text was changed to "Furthermore, during ASOND the flux coming from the IT site D (contained in area 2) divides into two, creating a more westward branching. In Figure 14-(b) the branching is visible in the ISW satellite measurements near latitude 4°N" (see pages 22-23, lines 383-385).**

18. Line 376. Mode-2 waves coming from D are mentioned here but in Figure 17 there is nothing coming from D.

**ANSWER:**

**We have added a green rectangle in Figure 7-(b) showing the waves that probably are coming from the D site. This information was added in the text as well. Please, see page 23, lines 286-287.**

---

## Author Comment (AC4)

**Spatial and temporal variability of mode-1 and mode-2 internal solitary waves from MODIS/TERRA sun glint off the Amazon shelf.**

Carina Regina de Macedo[1,2], Ariane Koch-Larrouy[2], Jose Carlos Bastos da Silva[3,4], Jorge Manuel Magalhães[3,5], Carlos Alessandre Domingos Lentini[6,7,8], Trung Kien Tran[1], Marcelo Caetano Barreto Rosa[7], Vincent Vantrepotte[1]

[1]Univ. Lille, CNRS, Univ. Littoral Côte d'Opale, IRD, UMR 8187 - LOG - Laboratoire d'Océanologie et de Géosciences, F-59000Lille, France.
[2]LEGOS, Université de Toulouse, CNES, CNRS, IRD, UPS, Toulouse, France.
[3]Department of Geosciences, Environment and Spatial Planning, Faculdade de Ciências da Universidade do Porto, Rua do Campo Alegre 687, 4169-007, Porto, Portugal.
[4]Instituto de Ciências da Terra, Polo Porto, Universidade do Porto, Rua do Campo Alegre 687, 4169-007, Porto, Portugal.
[5]CIIMAR, Universidade do Porto, Rua dos Bragas 289, 4050-123, Porto, Portugal.
[6]Department of Earth and Environment Physics, Physics Institute, Ondina Campus, Federal University of Bahia—UFBA, Salvador, Bahia, Brazil.
[7]Department of Oceanography, Geosciences Institute, Campus Ondina, Federal University of Bahia —UFBA, Salvador, Bahia, Brazil.
[8]Interdisciplinary Center for Energy and Environment (CIEnAm), Federal University of Bahia UFBA, Salvador, Bahia, Brazil.

**We thank the reviewer for taking the time to review our manuscript and especially for the important remark regarding the values of the diffusivities. In the following, our response to the reviewer's comment is in bold black color.**

The authors have addressed most of my comments and I think the paper is suitable for publication. My only remark concerns the values of the diffusivities, I think molecular diffusivities are not appropriate for geophysical flow it is really nearly equivalent to consider an inviscid flow. So putting such a low diffusivity does not really make sense it would be more straightforward to state that they consider inviscid solutions of the TG solution. If they really want to get an estimate of the diffusivities impact on TG solution they should consider eddy diffusivities which will be orders of magnitude larger.

**ANSWER:**
**The authors agree with the reviewer's comment. We changed the manuscript to state that we have considered inviscid solutions of the TGE. The appendix containing the method to solve the viscous TGE was excluded and all important information about the TGE is summarized in section 2.3: Theoretical calculation of IT velocities.**

---

## Referee Report (RR1)

*Review of*

**"Spatial and temporal variability of mode-1 and mode-2 internal solitary waves from MODIS/TERRA sun glint off the Amazon Shelf"**

*by C. R. de Macedo, A. Koch-Larrouy, J. C. B. da Silva, J. M. Magalhães,*
*C. A. D. Lentini, T. K. Tran, M. C. B. Roxa and V. Vantrepotte*

The paper has improved significantly. The writing is still a bit cumbersome and difficult to read in places.

**Comments**

1. In the first sentence of section 2.1 it is stated that 140 images were used. I suggest replacing all statements that 'more than a hundred' images were used with '140 images' throughout the manuscript. 'More than 100' is not precise enough. More than 100 could mean 101, 10,000 or even more.

2. The period MAMJJ includes spring and summer months but is referred to as 'spring' while ASOND includes summer and fall months and is referred to as summer/fall. Seems inconsistent not to refer to MAMJJ as spring/summer.

3. Line 15. "play a role in refracting the waves towards the northeast'. Or 'bending the waves ...". I'd delete the 'which gives them an extra offshore acceleration'. The meaning of this is not very clear.

4. Lines 29–30. Delete 'internal solitary waves' and just use the acronym as the acronym was defined on line 23.

5. Line 33. Delete 'generate'. The hotspots aren't generated. Internal waves are generated.

6. In the results section sub-patches are mentioned but not defined. Where in figure 2 are the subpatches? How were the distances between them, given in table 1, calculated?

7. Mention is made of wavelengths and velocities estimated from the images. The wording often implies that the propagation velocity is estimated independently of the wavelength (e.g., "mean propagation velocity/wavelength varies ...' on line 238) but in reality the velocity is simply the distance between consecutive wave packets divided by the $M_2$ tidal period. So if the wavelength increases by 10% then the velocity will as well. That the velocity is computed this way was not clear enough to me. It would help if the wording was changed to something like "... the wavelength increased by $x\%$ implying a corresponding increase in the propagation speed".

   Also, I think using the term 'wavelength' instead of 'inter-packet distance' is not a great idea.

8. Line 177. What does "calling for mode-1 waves" mean?

9. Line 182. What is the simulated mode-1 mean propagation velocity?

10. Lines 184–187. This seems a bit irrelevant and begs for an explanation of why the phase speed is proportional to the surface wave elevation and why it explains the higher underestimation of the mode-1 waves.

11. Lines 197. Here it is stated that the mode-1 velocities are underestimated by 22%. What about the mode-2 velocities?

12. Line 201. What is meant by 'joining area 2'? I do not see a green rectangle in Figure 7(b).

13. Line 227. Simplify: "... waves travel in a more eastward direction"

14. Line 238. Why are higher wavelengths associated with neap tides rather than spring tides where the stronger tides would suggest larger ITs being generated with larger propagation speeds.

15. Line 344. Should 'Area A' be 'Area 2'?

16. Lines 355–356. What is meant by 'refracting the waves northeast'. It seems like everything is propagating roughly northeast. Do you mean more eastward or more northward?

17. Lines 373–374. Mention is a northwest branch is made but I can't see anything propagating northwest in any of the Figures.

18. Line 376. Mode-2 waves coming from D are mentioned here but in Figure 17 there is nothing coming from D.